# Grokking of Implicit Reasoning in Transformers: A Mechanistic Journey to the Edge of Generalization

**Boshi Wang**♠    **Xiang Yue**◇*    **Yu Su**♠    **Huan Sun**♠
♠The Ohio State University        ◇Carnegie Mellon University
{wang.13930,yue.149,su.809,sun.397}@osu.edu

## Abstract

We study whether transformers can learn to *implicitly* reason over parametric knowledge, a skill that even the most capable language models struggle with. Focusing on two representative reasoning types, composition and comparison, we consistently find that transformers *can* learn implicit reasoning, but only through *grokking*, i.e., extended training far beyond overfitting. The levels of generalization also vary across reasoning types: when faced with out-of-distribution examples, transformers fail to systematically generalize for composition but succeed for comparison. We delve into the model's internals throughout training, conducting analytical experiments that reveal: 1) the mechanism behind grokking, such as the formation of the generalizing circuit and its relation to the relative efficiency of generalizing and memorizing circuits, and 2) the connection between systematicity and the configuration of the generalizing circuit. Our findings guide data and training setup to better induce implicit reasoning and suggest potential improvements to the transformer architecture, such as encouraging cross-layer knowledge sharing. Furthermore, we demonstrate that for a challenging reasoning task with a large search space, GPT-4-Turbo and Gemini-1.5-Pro based on non-parametric memory fail badly regardless of prompting styles or retrieval augmentation, while a fully grokked transformer can achieve near-perfect accuracy, showcasing the power of parametric memory for complex reasoning.[2]

## 1 Introduction

Large language models (LLMs) have been shown deficient in *implicit* reasoning with their parametric memory of knowledge and rules. For example, a range of LLMs are found to be incapable of robustly composing internalized facts [48, 71], and even GPT-4 [42] cannot adequately compare entities' attributes despite knowing them [1].

Deficiency in implicit reasoning has profound implications. It implies the models' limitations in inducing structured and compressed representations of facts and rules, which lead to redundant knowledge storage and difficulty in propagating knowledge updates [76], and importantly, fundamentally impede the model from *systematic generalization* over knowledge [25]. While explicit verbalizations of reasoning steps (e.g., chain-of-thought rationales) can improve task performance [67, 64, 73, 55, 31], they are not available during large-scale (pre-)training where the model's core capabilities are acquired [77, 29].

*Is implicit reasoning doomed given that even the most capable models struggle? Can it be resolved by further scaling data and compute, or are there fundamental limitations of the transformer [62] that prohibit robust acquisition of this skill?*

---

*Project started when at OSU.

[2]Code and data: https://github.com/OSU-NLP-Group/GrokkedTransformer.

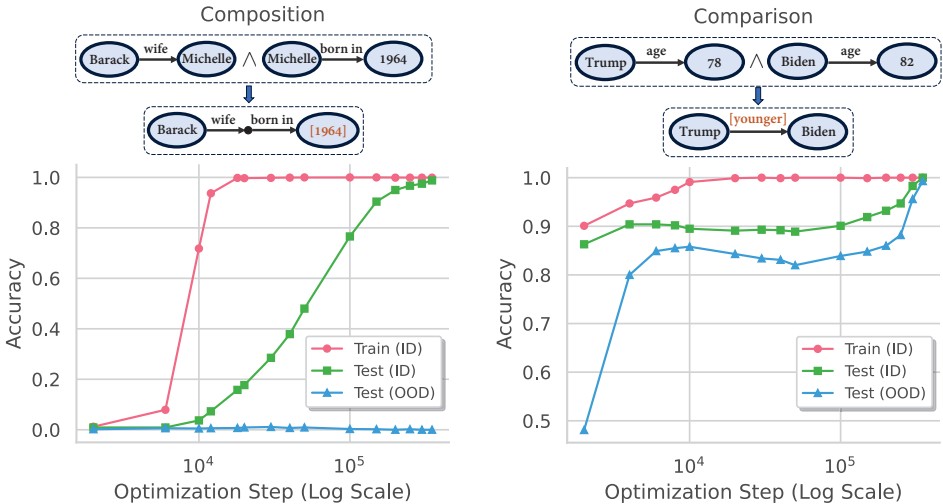

Figure 1: We find that transformers can learn to reason implicitly, but this skill is only robustly acquired through *grokking*, i.e., an extended period of training far beyond overfitting. Moreover, the transformer fails to *systematically* generalize for composition, yet succeeds for comparison. We conduct a mechanistic study into the model internals throughout grokking, which reveals distinct generalizing circuits across the two tasks (Figure 4, 5) that explains the variations in systematicity.

In this paper, we rigorously study these questions by constructing synthetic training and evaluation datasets, training transformers from scratch, and examining their generalization. We conceptualize reasoning as the *induction and application of inference rules*, and expose the model to a mixture of "atomic facts" and "inferred facts" (which are deduced from the atomic facts via a set of latent rules), resembling "axioms" and "theorems" in a formal system. To evaluate how well the model learns the rules, we test its ability to make novel deductions (i.e., completing unseen inferred facts) in both in-distribution (ID) and out-of-distribution (OOD) scenarios.[3] This approach allows us to control the training data and perform clean evaluations, which would be challenging when studying existing LLMs trained on uncontrolled data.

Our experiments reveal that transformers *can* learn to perform implicit reasoning, but this skill is only robustly acquired through *extended training far beyond overfitting* (Figure 1), a phenomenon known as *grokking* [47]. We find that the speed of improvement in generalization correlates with the *ratio* between inferred and atomic facts in training, and depends little on the absolute *size* of the training data (Figure 2). This suggests a correction of prior explanations of grokking based on *critical data size* [33, 61, 78, 21], in that it should instead be the *critical data distribution* that decides the characteristics of grokking. Our findings extend prior observations of the grokking phenomenon primarily in algorithmic and linguistic tasks [47, 38] to the domain of knowledge-based reasoning, and deepen our understanding of the grokking phenomenon.

Moreover, we find that the transformer exhibits different levels of systematicity across reasoning types. While ID generalization is consistently observed, in the OOD setting, the model fails to systematically generalize for composition but succeeds in comparison (Figure 1). To understand why this happens, we conduct mechanistic analysis of the internal mechanisms of the model. The analysis uncovers the gradual formation of the *generalizing circuit* throughout grokking and establishes the connection between systematicity and its configuration, specifically, the way atomic knowledge and rules are stored and applied within the circuit. Our findings imply that proper cross-layer memory-sharing mechanisms for transformers such as memory-augmentation [54, 17] and explicit recurrence [7, 22, 57] are needed to further unlock transformer's generalization.

Finally, to demonstrate the power and potential of parametric memory for complex reasoning, we show that for a reasoning task with a large search space, a fully grokked transformer can achieve near-perfect accuracy, while state-of-the-art LLMs like GPT-4-Turbo [43] and Gemini-1.5-Pro [16] based on non-parametric memory fail badly regardless of prompting styles or retrieval augmentation.

---

[3]Definitions of ID/OOD are introduced in §2.

## 2 General Setup

**Training data & ID/OOD evaluation.** As stated in §1, we are interested in whether transformers can induce and apply latent rules over knowledge implicitly in a generalizable way. We create a data-generating process consisting of 1) sampling a set of basic *atomic facts*, and 2) using the atomic facts and latent rules to deduce *inferred facts*. To better characterize the level of generalization acquired by the model, we evaluate the model's in-distribution (ID) and out-of-distribution (OOD) performance. We prepare two separate sets of atomic facts: $\texttt{atomic}_\texttt{ID}$ and $\texttt{atomic}_\texttt{OOD}$. Our training set includes *all* the atomic facts and a uniformly random portion of the inferred facts deduced from $\texttt{atomic}_\texttt{ID}$, which we call $\texttt{train\_inferred}_\texttt{ID}$. For evaluation, (1) ID generalization aims to evaluate whether the model learns the latent rules correctly, by testing its ability to complete unseen inferred facts also deduced from $\texttt{atomic}_\texttt{ID}$, which we denote by $\texttt{test\_inferred}_\texttt{ID}$. (2) OOD generalization aims to evaluate the *systematicity* [25] acquired by the model, namely, the ability to apply rules over knowledge regardless of its distribution. To do this, we test the model on the facts deduced from $\texttt{atomic}_\texttt{OOD}$, denoted by $\texttt{test\_inferred}_\texttt{OOD}$.

**Model & optimization.** We use a standard decoder-only transformer model as in GPT-2 [50] with 8 layers, 768 hidden dimensions and 12 attention heads (we explore the impact of different model scales in Appendix B). Optimization is done by AdamW [34] with learning rate $10^{-4}$, batch size 512, weight decay 0.1 and 2000 warm-up steps. Notably, models are trained for a large number of epochs/steps beyond the point where training performance saturates. More details are in Appendix A.

## 3 Composition—Delayed Generalization without Systematicity

We begin our investigation with *composition*, where a model needs to "chain" different pieces of facts, e.g., *"Barack's wife is Michelle"* and *"Michelle is born in 1964"*, to successfully complete a compositional sentence, e.g., *"Barack's wife is born in [1964]"*. Prior work extensively studied whether transformer-based language models can perform implicit composition, and negative results are consistently reported [48, 1, 71]. Specifically, there exists a *"compositionality gap"* [48], i.e., the frequency at which the model knows all the underlying basic facts but fails to compose them, which is considerable across different LLMs and does not decrease as models scale. Are transformers doomed to fail on such kind of reasoning, and if so, why?

### 3.1 Setup

We focus on two-hop composition in this work. For atomic facts, we generate a random knowledge graph $\mathcal{G}$ consisting of $|\mathcal{E}|$ entities and $|\mathcal{R}| = 200$ relations, where each entity (as the subject) has 20 random distinct relations that each connects to another random entity (as the object). The atomic facts are then the edges, i.e., *(subject, relation, object)* triplets in $\mathcal{G}$, which we partition disjointly into $\texttt{atomic}_\texttt{ID}$ and $\texttt{atomic}_\texttt{OOD}$ (95%: 5%). The rule of (two-hop) composition is

$$\forall h, b, t \in \mathcal{E}, \forall r_1, r_2 \in \mathcal{R}, (h, r_1, b) \wedge (b, r_2, t) \implies (h, r_1, r_2, t), \tag{1}$$

which is used to deduce the ID and OOD inferred facts from $\texttt{atomic}_\texttt{ID}$ and $\texttt{atomic}_\texttt{OOD}$, respectively. For convenience, in the above rule, we will call $h$ the *head* entity, $b$ the *bridge* entity, and $t$ the *tail* entity. For both atomic and inferred facts, training/testing is done by having the model predict the final tail entity. We assign a unique token to each relation/entity by default, and also find that the results are robust to different tokenizations (details in Appendix C).

We study the influence of the following two aspects on the model's learned behaviors:

- **Ratio between inferred and atomic facts.** By varying the amount of inferred facts included in training, we study the effect of the ratio $\phi = |\texttt{train\_inferred}_\texttt{ID}|/|\texttt{atomic}_\texttt{ID}|$ on the model.
- **Training data size.** We study the impact of the training data size by varying $|\mathcal{E}|$, the total number of entities, while controlling the ratio $\phi$. Note that the size of training data (both atomic/inferred facts) scales linearly with $|\mathcal{E}|$.

### 3.2 Results

**Grokking observed in ID generalization but not in OOD generalization**. Figure 1(left) shows the model's accuracy on the train and test facts throughout optimization, with $|\mathcal{E}| = 2000$ and $\phi = 7.2$.

We find that the model *can* generalize to ID test examples, but high performance is only achieved through extended training far beyond overfitting, a phenomenon called *grokking* [47]. Specifically, the training performance saturates (over 99% accuracy on both atomic and inferred facts) at around 14K optimization steps, before which the highest ID generalization accuracy is merely 9.2%. However, generalization keeps improving by simply training for longer, and approaches almost perfect accuracy after extended optimization lasting around 50 times the steps taken to fit the training data. On the other hand, OOD generalization is never observed. We extend the training to 2 million optimization steps, and there is still no sign of OOD generalization.

**Inferred/atomic ratio $\phi$ correlates with generalization speed**. Figure 2(a) shows the ID test accuracy across different $\phi$. We omit the other splits since for all settings, the training performance saturates quickly and the OOD test accuracy remains at zero as earlier.[4] It could be seen that the ratio $\phi$ strongly correlates with the *speed* of generalization. A very large ratio can push generalization to improve at a similar pace as the model fits the training data, reducing the need for extended training.[5]

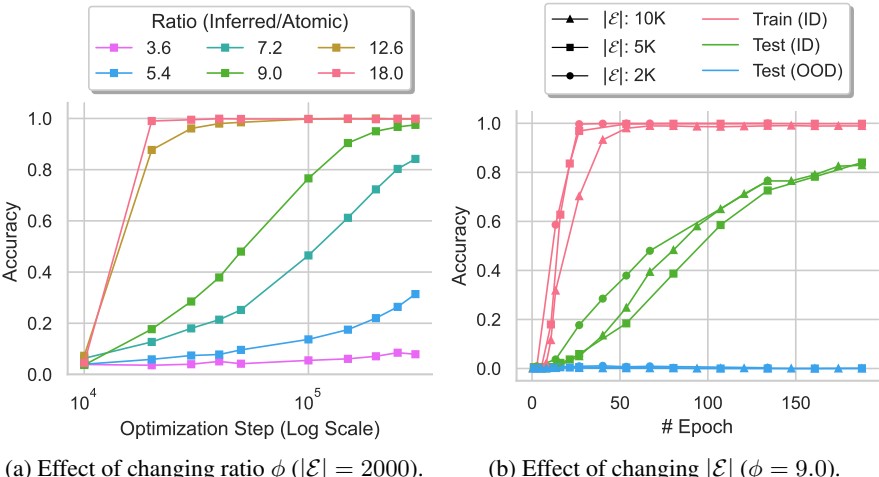

(a) Effect of changing ratio $\phi$ ($|\mathcal{E}| = 2000$).  (b) Effect of changing $|\mathcal{E}|$ ($\phi = 9.0$).

Figure 2: The speed of grokking on the in-distribution (ID) test performance (a) correlates with the *ratio* between inferred and atomic facts, and (b) is not influenced by the *size* of training data.

**Training data *distribution*, instead of training data size, qualitatively influences generalization behavior**. When $\phi$ increases and $|\mathcal{E}|$ holds constant, the *size* of training data also gets larger. Prior studies hypothesize that training data size plays a central role in order for grokking to happen. In particular, previous work connects grokking with the notion of *critical data size* (CDS) [33, 61, 78, 21], where it is hypothesized that CDS marks the shift from memorization to generalization (via grokking), and the speed of generalization improves as the training data further scales. However, results from our controlled experiments seem to contradict such a hypothesis. Figure 2(b) shows the results of varying $|\mathcal{E}|$ with a fixed $\phi = 9.0$, where we change the horizontal axis from optimization step to epoch for better visualization.[6] When fixing the ratio $\phi$, the training data size does *not* qualitatively affect the model's generalization. Specifically, scaling the data affects neither the relative speed of ID generalization and training improvement (as seen by the rather constant "gap" between train_inferred$_{\text{ID}}$ and test_inferred$_{\text{ID}}$ curves), nor the systematicity level (OOD performance stays zero). We also run the experiments across different $\phi$ and find the results to be consistent. This suggests that *critical data "distribution", not size, may be the actual deciding factor behind grokking and generalization.* In addition, we find that scaling up the model size also does not qualitatively change the generalization behaviors observed here (Appendix B), and the main pattern is that larger models converge in fewer optimization steps, which shares with prior findings [60, 28].

**Summary.** We have shown that transformers are capable of acquiring the rule of composition through grokking, with controlled experiments suggesting the crucial factor of data distribution (e.g., the inferred/atomic ratio $\phi$) in characterizing the model's generalization. However, important questions

---

[4]The training performances of all settings saturate within 25K steps, where larger $\phi$ takes more steps.

[5]When $\phi = 18.0$, the model achieves 96.7% accuracy before training performance saturates.

[6]The optimization steps for each epoch scale linearly with the training size since we use a fixed batch size.

still remain: *what happens during grokking, why does it happen, and why do transformers struggle with OOD examples?* Answering these questions requires a deeper understanding of (the changes in) the model's inner workings, which we investigate next.

### 3.3 Analyzing the inner workings of the model throughout grokking

We analyze the internal mechanisms within the model via a combination of two prevalent approaches: logit lens and causal tracing. We apply our analysis to the setting with $|\mathcal{E}| = 2000, \phi = 9.0$ on 300 random examples from `train_inferred`$_{\text{ID}}$.

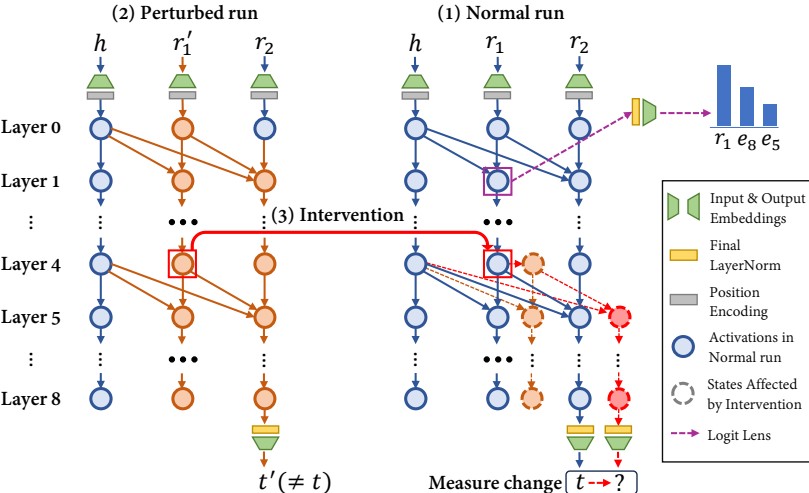

Figure 3: Illustration of our circuit analysis approach (on the composition task). We use logit lens to interpret individual states, and use causal tracing to measure the strength of connections between states. Details are in the main content.

**Logit lens**. We interpret individual hidden states via logit lens [40, 15, 71], where the activation is converted into a set of logits for each vocabulary token by multiplying with the output embedding matrix. We follow the recent practice [71] where the activation first goes through the transformer's final normalization layer before multiplying with the output embedding (Figure 3, top right).

**Causal tracing**. The transformer could be viewed as a causal graph [46] that propagates information from the input to the output through a grid of intermediate states, which allows for a variety of causal analyses on its internal computations [63, 35, 19, 65, 12]. For convenience, we will refer to a hidden state by $S[i, a]$, where $i$ is the layer index and $a$ marks the role of the input token at the same position as the state (one of $\{h, r_1, r_2\}$). We illustrate our method in Figure 3, where the hidden state of interest is $S[4, r_1]$ and the target is the model's prediction state $S[8, r_2]$. There are in total three steps:

1. The **normal run** records the model's hidden state activations on a regular input $(h, r_1, r_2)$. Note that since the model maintains perfect training performance throughout grokking, the final prediction is always the ground truth tail entity $t$.[7]
2. In the **perturbed run**, a slightly perturbed input is fed to the model which changes the prediction, where again the hidden state activations are recorded. For the perturbation, prior work has explored adding noise to the input [35] and replacing key tokens with semantically close ones [63, 12]. We adopt token replacement which avoids unnecessary distribution shifts [74]. Specifically, for the hidden state of interest, we replace the input token at the same position as the state to be a random alternative of the same type (e.g., $r_1 \to r_1'$) that leads to a different target prediction (e.g., $t \to t'$).
3. **Intervention**. During the normal run, we intervene the state of interest by replacing its activation with its activation in the perturbed run. We then run the remaining computations and measure if the target state (top-1 token through logit lens) is altered. The ratio of such alterations (between 0 and 1) among the examples quantitatively characterizes the *causal strength* between the state of interest and the target.

---

[7]For convenience, when we refer to a state as a token, we mean the top token of the state via logit lens.

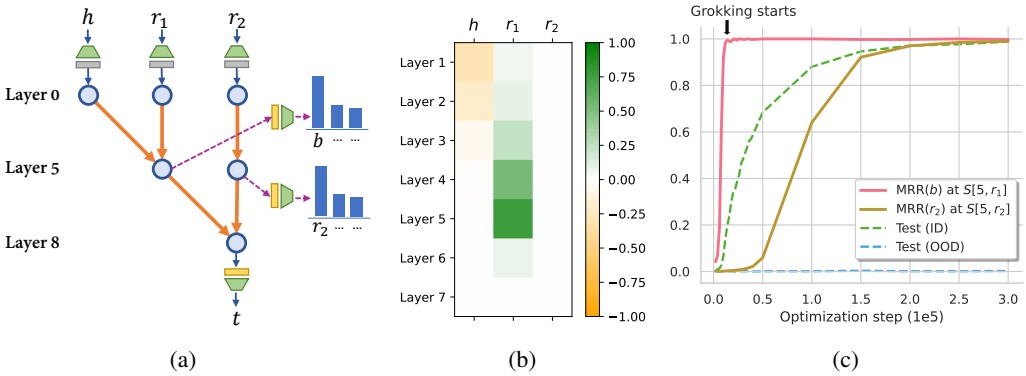

(a)  (b)  (c)

Figure 4: The (evolution of) generalizing circuit for composition. (a) The generalizing circuit. (b) The *change* in causal strengths during grokking, where the target is the prediction state. (c) Mean reciprocal rank (via logit lens) of the bridge entity $b$ at $S[5, r_1]$ and second relation $r_2$ at $S[5, r_2]$.

**The generalizing circuit**. We run a set of causal tracing and logit lens experiments across different model checkpoints throughout training. The discovered generalizing circuit (i.e., the causal computational pathways after grokking) is illustrated in Figure 4(a). Specifically, we locate a highly interpretable causal graph consisting of states in layer $0$, $5$, and $8$, where we have pruned away the weak nodes/connections (details in Appendix D). Layer $5$ splits the circuit into lower and upper layers, where 1) the lower layers retrieve the first-hop fact $(h, r_1, b)$ from the input $h, r_1$, store the bridge entity $b$ in $S[5, r_1]$, and "delay" the processing of $r_2$ to $S[5, r_2]$; 2) the upper layers retrieve the second-hop fact $(b, r_2, t)$ from $S[5, r_1]$ and $S[5, r_2]$, and store the tail $t$ to the output state $S[8, r_2]$.

**What happens during grokking?** To understand the underlying mechanism behind grokking, we track the strengths of causal connections and results from logit lens across different model checkpoints during grokking (the "start" of grokking is the point when training performance saturates). We observe two notable amplifications (within the identified graph) that happen during grokking. The first is the causal connection between $S[5, r_1]$ and the final prediction $t$, which is very weak before grokking (Appendix D) and grows significantly during grokking (Figure 4(b)). The second is the $r_2$ component of $S[5, r_2]$ via logit lens, for which we plot its mean reciprocal rank (MRR) (Figure 4(c)). Additionally, we find that the state $S[5, r_1]$ has a large component of the bridge entity $b$ throughout grokking (Figure 4(c)). These observations strongly suggest that the model is *gradually forming the second hop in the upper layers (5-8) during grokking*. This also indicates that, before grokking, the model is very likely mostly memorizing the examples in `train_inferred`$_\text{ID}$ by *directly associating $(h, r_1, r_2)$ with $t$*, without going through the first hop.

**Why does grokking happen?** These observations suggest a natural explanation of why grokking happens through the lens of circuit efficiency [61]. Specifically, as illustrated above, there exist both a memorizing circuit $C_{mem}$ and a generalizing circuit $C_{gen}$ that can fit the training data. While $C_{mem}$ is learned first (which causes training performance to saturate quickly), $C_{gen}$ is relatively more *efficient*, in the sense that it could fit the data with a lower complexity. To see this, we can compare the amount of facts $C_{mem}$ and $C_{gen}$ need to store (denoted as $N_{mem}$ and $N_{gen}$) as a proxy for their complexity.[8] $C_{mem}$ stores both atomic facts and inferred facts in the weights. $C_{gen}$ (Figure 4(a)) stores the atomic facts in the lower layers, and another copy of the atomic facts that *appear as the second hop in the inferred facts* in the upper layers. As the inferred/atomic ratio $\phi$ increases, $N_{mem}$ would increase rapidly while $N_{gen}$ increases slowly and is always bounded by two times the total amount of atomic facts, and hence, the relative efficiency of $C_{gen}$ increases. In the long run, the model will be incentivized to transition from $C_{mem}$ to $C_{gen}$ due to implicit bias of the optimization [53] and explicit regularization such as weight decay which prefers more efficient circuits, and the transition would happen faster as $\phi$ increases. This also explains why the training data size does not affect the speed of grokking, since *solely increasing the size does not change the relative efficiency of $C_{mem}$ and $C_{gen}$*. The explanation also implies that a larger regularization factor should accelerate grokking (and vice versa), which we confirm by varying the degree of weight decay (Appendix E.1).

---

[8]While the circuits also consist of other components, they pale in comparison as the number of facts scales.

**Explaining and mitigating the deficiency in OOD generalization.** The configuration of $C_{gen}$ also has another important implication: while the model does acquire compositionality through grokking, it *does not have any incentive to store atomic facts in the upper layers that do not appear as the second hop during training*. This explains why the model fails in the OOD setting where facts are only observed in the atomic form, not in the compositional form—the OOD atomic facts are simply not stored in the upper layers when queried during the second hop.[9] Such issue originates from the non-recurrent design of the transformer architecture which forbids memory sharing across different layers. Our study provides a mechanistic understanding of existing findings that transformers seem to reduce compositional reasoning to linearized pattern matching [10], and also provides a potential explanation for the observations in recent findings that LLMs only show substantial positive evidence in performing the first hop reasoning but not the second [71]. Our findings imply that proper cross-layer memory-sharing mechanisms for transformers such as memory-augmentation [54, 17] and explicit recurrence [7, 22, 57] are needed to improve their generalization. We also show that a variant of the parameter-sharing scheme in Univeral Transformer [7] can improve OOD generalization in composition (Appendix E.2).

## 4 Comparison—Systematic Generalization via Parallel Circuit

We have just shown that the vanilla transformer fails to achieve OOD generalization for composition, *but is the vanilla transformer generally incapable of acquiring systematic implicit reasoning skills?* We show that for *comparison*, a task where SoTA LLMs such as GPT-4 also struggle [1], the vanilla transformer *does* have the capability to acquire systematic generalization, again through grokking. On the surface, it seems that the comparison task is no different than the composition task—both require retrieving and reasoning over two pieces of facts. However, as it turns out through our analysis, *the comparison task emits a "parallel circuit" that is learned by the transformer during grokking, which allows atomic facts to be stored and retrieved in the same region and enables systematicity to happen.*

**Setup.** The comparison task involves comparing the attribute values of entities. We assume there are $|\mathcal{E}| = 1000$ entities, $|\mathcal{A}| = 20$ attributes and $|\mathcal{V}| = 20$ ordinal values for the attributes. Each attribute $a \in \mathcal{A}$ has a label space $\{a_<, a_=, a_>\}$, a set of relations specifying its comparative form. For example, an attribute *age* would have $a_<, a_=, a_>$ to be *younger, contemporary, older*, respectively.

The atomic facts are *(entity, attribute, value)* triplets, where we assign a random value $v \in \mathcal{V}$ for each $(e, a) \in \mathcal{E} \times \mathcal{A}$. Again, we randomly partition the atomic facts into `atomic_ID` and `atomic_OOD` (90%: 10%). The rules of comparison are:

$$\forall e_1, e_2 \in \mathcal{E}, \forall a \in \mathcal{A}, \forall v_1, v_2 \in \mathcal{V}, (e_1, a, v_1) \wedge (e_2, a, v_2) \wedge v_1 < v_2 \implies (a, e_1, e_2, a_<),$$
$$\forall e_1, e_2 \in \mathcal{E}, \forall a \in \mathcal{A}, \forall v_1, v_2 \in \mathcal{V}, (e_1, a, v_1) \wedge (e_2, a, v_2) \wedge v_1 = v_2 \implies (a, e_1, e_2, a_=), \quad (2)$$
$$\forall e_1, e_2 \in \mathcal{E}, \forall a \in \mathcal{A}, \forall v_1, v_2 \in \mathcal{V}, (e_1, a, v_1) \wedge (e_2, a, v_2) \wedge v_1 > v_2 \implies (a, e_1, e_2, a_>).$$

Take the attribute *age* as an example, the first rule means if the *age* of $e_1$ is *smaller than* the *age* of $e_2$, then we can infer "In terms of *age*, the relation between $e_1$ and $e_2$ is *younger*". Each entity/attribute/value/label is assigned a unique token, and training/testing is done by having the model predict the last token (attribute value for atomic facts; comparative relation for inferred facts).

**Results & analysis**. Figure 1(right) shows the results for $\phi = 7.2$, and we include more results in Appendix E.3. It can be seen that 1) the model again acquires robust generalization only through grokking; 2) surprisingly, the model also achieves systematicity in generalization, different from the case of composition.

Analyzing the model's internals similarly as in §3.3 (details in Appendix D), we find the generalizing circuit for comparison illustrated in Figure 5(a). On a separate stream, the model prepares the label space $\{a_<, a_=, a_>\}$ from $a$ and stores it in $S[7, a]$. In the lower layers (0-5), the model retrieves the two atomic facts and stores the attribute values $v_1$ and $v_2$ at $S[5, e_1]$ and $S[5, e_2]$. Then, the upper layers (5-8) compare $v_1, v_2$ and fetch the label from $S[7, a]$ based on the comparison result. Importantly, there is a major difference compared with the circuit for composition: the two atomic facts are retrieved *in parallel*, which suggests that the atomic facts are stored solely in the lower layers, without having separate copies across different regions as in the circuit for composition. This explains why systematicity could happen: OOD facts are now stored and accessed in the same way as ID

---

[9]We verified that in the OOD setting, $S[5, r_1]$ and $S[5, r_2]$ encode $b$ and $r_2$ respectively as in the ID case.

facts. Tracking the changes in the model throughout grokking, we observe significantly strengthened causal connections from $S[7,a]$ and $S[5,e_1]$ to the final prediction (Figure 5(b)). We also find that throughout grokking, $S[7,a]$ always encodes the label space and $S[5,e_1], S[5,e_2]$ gradually encode the two attribute values (Figure 5(c)). This confirms that a similar transition from $C_{mem}$ to $C_{gen}$ happens during grokking.

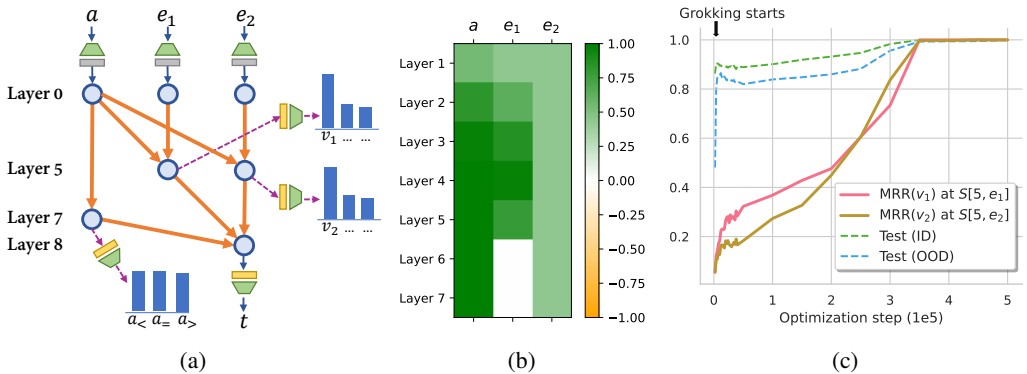

(a)            (b)            (c)

Figure 5: The (evolution of) generalizing circuit for comparison. (a) The generalizing circuit. (b) The *change* in causal strengths during grokking, where the target is the prediction state. (c) Mean reciprocal rank (via logit lens) of the two attribute values ($v_1, v_2$) at $S[5,e_1]$ and $S[5,e_2]$.

The findings here showcase transformer's ability to learn parallel solutions to seemingly sequential problems, akin to the findings in Liu et al. [30] where it is shown that transformers can learn "shortcuts" to automata. The difference in the acquired generalization across the two tasks that we study also emphasizes the need for *controlled and mechanistic* study on understanding the transformer's reasoning before making general claims on its limitations.

## 5 The Power of Parametric Memory for Complex Reasoning

At the high level, our study so far paves the way towards better understanding and improving transformer's reasoning with *parametric* representation of knowledge and rules. But why is parametric memory practically important? Can we not simply enhance LLMs with non-parametric memory, e.g., by using their long-context modes and/or doing explicit retrieval, to solve the tasks at hand?

We believe parametric memory has its unique capability to perform *deep compression and integration of information* for complex reasoning. To showcase the potential of parametric memory for complex reasoning, we create a difficult reasoning task with a *large search space*, and show that 1) it is far out of reach even for current SoTA models (e.g., GPT-4-Turbo [43] and Gemini-Pro-1.5 [16]) based on non-parametric memory; 2) a fully grokked transformer can solve the task with near-perfect accuracy.

Our task is a variation of the comparison task above where we use a simple way to massively expand the search space, based on an additional set of rules that are already contained within the task itself, namely, the *(anti-)symmetry* and *transitivity* of comparison:

$$\forall e_1, e_2 \in \mathcal{E}, \forall a \in \mathcal{A}, (a, e_1, e_2, a_</a_=/a_>) \implies (a, e_2, e_1, a_>/a_=/a_<),$$
$$\forall e_1, e_2, e_3 \in \mathcal{E}, \forall a \in \mathcal{A}, \forall y \in \{a_<, a_=, a_>\}, (a, e_1, e_2, y) \land (a, e_2, e_3, y) \implies (a, e_1, e_3, y). \quad (3)$$

In the original setting (§4) of the task, for the OOD test set, one can simply retrieve the two OOD facts and compare the attribute values, which requires no further search. We change the task setting via the following. For each attribute, 1) we do *not* add the OOD atomic facts into training, and 2) we add a random portion of the comparisons between ID entities and OOD entities into training. We test the models on queries consisting of *derivable* (from all training facts) comparisons between OOD entities where any possible proof would involve rules from *both* Eqs.(2) and Eqs.(3). Consequently, answering a test query would require the model to successfully *locate two ID bridge entities* which can connect the two query entities into a proof (Figure 6). We select a balanced (by $a_<, a_=, a_>$) subset from these queries for evaluation. More details are included in Appendix F.

The difficulty of such a task is two-fold. First, the search space is large. For example, on average, each query entity connects with more than $50$ facts, and each bridge entity in the ground truth proof

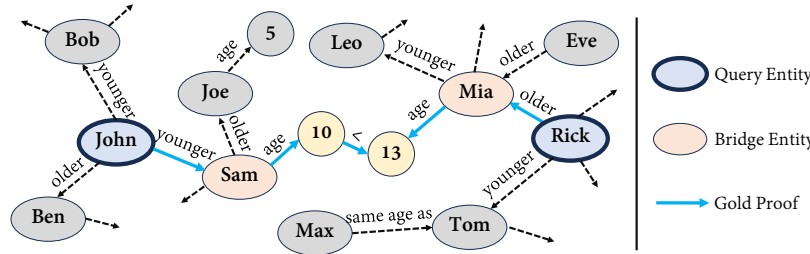

Figure 6: Illustration of the complex reasoning task, which involves comparing the attributes of two query entities based on a set of facts encompassing a large search space.

connects with more than 900 facts. Second, there are *no surface form clues* to exploit and bias the search towards the ground truth proof, unlike most conventional QA benchmarks where the proof steps are transparent from the query.

To test LLMs based on non-parametric memory, we translate the facts into natural language by simple templates (Appendix F). Facts/queries for each attribute are grouped/tested separately.[10] We test both the vanilla setup where all facts (28.2K on average) are loaded into the LLM context, and the retrieval-augmented setup (5.4K facts retrieved on average) where the two-hop neighborhoods of the two query entities are retrieved, which includes enough facts to deduce the answer. We also try both standard prompting where the model answers directly, and chain-of-thought (CoT) prompting where the model is prompted to verbalize the reasoning. We test GPT-4-Turbo and Gemini-Pro-1.5, where for GPT-4-Turbo we only test the retrieval-augmented setup due to context length limit.

Table 1: Results on the complex reasoning task. Direct/CoT: predict the answer directly/verbalize the reasoning steps. "+R": retrieval augmentation.

| | GPT-4-Turbo | | Gemini-Pro-1.5 | | | | Grokked Transformer |
| --- | --- | --- | --- | --- | --- | --- | --- |
| | Direct+R | CoT+R | Direct | CoT | Direct+R | CoT+R | |
| **Accuracy (%)** | 33.3 | 31.3 | 28.7 | 11.3 | 37.3 | 12.0 | **99.3** |

**Results**. As shown in Table 1, all models based on non-parametric memory fail badly, where the only setting that surpasses random guess (33.3%) is the retrieval-augmented setting with Gemini-Pro-1.5 and direct answer prediction. Intriguingly, LLMs perform worse (especially Gemini) when prompted to reason verbally. Through closer examinations, we find that with CoT, 70.7% of Gemini's responses conclude that the answer cannot be decided (which we treat as wrong since the answer can be decided) after a series of search steps.[11] More shockingly, *most of the CoT rationales that achieve the correct final answer are actually wrong due to either hallucinating underivable facts or logical errors*. This illustrates the current models' inability to reason deeply with non-parametric memory. On the other hand, a grokked transformer, which is trained extensively on the given facts to compress and integrate the information to the extreme, could achieve near-perfect accuracy. By examining the model, we find that it acquires the same generalizing circuit as in Figure 5(a), and remarkably, even though not explicitly encouraged/trained to do this, the model successfully infers most of the OOD entities' attribute values by integrating the observed training facts (Appendix F).

## 6 Related Work

**Knowledge and reasoning in language models**. Numerous work finds that transformer language models, even SoTA ones such as GPT-4, struggle in implicit reasoning over their parametric knowledge [56, 23, 52, 51, 48, 1, 71], suggesting their limitations in inducing structured and compressed representations of facts and rules during training. A series of efforts try to understand transformer's knowledge and reasoning through controlled experiments [24, 49, 10, 66], which is also our focus. We

---

[10]This could also be thought of as performing a retrieval step into the memory based on the attribute.

[11]The ratio drops a bit to 58.7% when augmenting with retrieval.

find that transformers can learn implicit reasoning over knowledge through grokking, and characterize the connection between the acquired systematicity level and the inductive bias of transformer.

**"Chain-of-Thought" and verbalized reasoning**. A series of studies prompt/fine-tune language models to verbalize (i.e., generate) the intermediate knowledge and reasoning steps [67, 64, 73, 55, 31, 72] during inference, which has been shown to improve performance especially for large models with strong generation capabilities. There are also theoretical results showing the benefits of such verbalizations [11, 27]. Our focus here on implicit reasoning is orthogonal, and it is an interesting open problem to have principled understandings of the role of such verbalizations in reasoning problems, and also develop methods that can decide the appropriate balance between implicit and explicit reasoning to handle challenging problems with large intrinsic complexity. Relatedly, recent work also finds that explicit verbalizations could be a useful medium for teaching models to reason implicitly via distillation or curriculum [9, 8].

**Grokking** is first discovered by Power et al. [47] on a set of small algorithmic reasoning tasks. The intriguing phenomenon inspired follow-up works proposing different explanations and expanding the set of tasks where grokking is observed [59, 32, 6, 41, 39, 61, 36, 38, 33, 78, 21]. To our knowledge, we are the first work to observe grokking in the domain of knowledge-based reasoning, and our controlled experiments suggest potential corrections of prior hypotheses based on critical data size. Our formulation of rule induction from atomic and inferred facts is general, which we hope could inspire future work on understanding grokking and generalization in deep learning.

**Analyzing the inner workings of neural models**. Recent work tries to open up the "black box" of neural models through a wide range of techniques; see survey in Ferrando et al. [13]. We apply causal tracing [63, 35, 19, 65, 12] and logit lens [40, 15, 71] to discover interpretable circuits in the model to understand the grokking process and how/why generalization happens.

**Parametric and non-parametric memory**. Our focus in this work is on parametric memory in language models, and an orthogonal direction is to enhance models with non-parametric memory, such as extending the effective context length [70, 43, 14, 16] and augmenting with retrieval [18, 26, 3, 58, 75, 37]. The two types of memory are largely complementary to each other—parametric memory has its unique ability to compress and integrate information but is also inevitably lossy and subject to hallucination, while non-parametric memory is lossless and could also provide attribution. Similarly for humans—a human acquires expertise in a domain by acquiring and structuring knowledge in the brain (parametric), but he/she also wouldn't memorize all pieces of details and could refer to the source when necessary (non-parametric). How to decide the tradeoff between parametric and non-parametric memory (or, how to define the objective for such a tradeoff) is another interesting open problem for future work.

## 7    Conclusion

We find that transformers are capable of learning to implicitly reason over parametric knowledge, however, such a skill is only robustly acquired through extended training far beyond the point of overfitting, or *grokking*. Mechanistic analysis into the model's internals reveals the configuration and gradual formation of the generalizing circuit, and also explains the different levels of systematicity the model acquires across tasks. These findings guide data and training setup to better induce implicit reasoning, and suggest potential improvements to the transformer architecture to further unlock its generalization. We conclude by showcasing the unique power of parametric memory on a challenging reasoning task with a large search space.

## Limitations

**Scope of our task formulation**. As mentioned in §1, we formulate the implicit reasoning problem as induction and application of inference rules from a mixture of atomic and inferred facts. This may not apply to the full spectrum of reasoning which has a range of different types and meanings [20]. Still, our formulation could capture the nature of a wide range of reasoning problems, and crucially, we believe that it is a good conceptualization of certain aspects of language model (pre-)training, where the model needs to both memorize the "atomic" world knowledge and also induce generalizable rules from the massive amount of records of human activities, which allow the model to connect and reason over knowledge, and ultimately help with humans.

**Abstract nature and connection with practice**. Our study has an abstract nature, which is required for the complete solidity and rigor of our experiments and evaluation. We also strive to make sure that the results are robust to different setups closer to practice through additional experiments (Appendix B,C,E). Still, there are certain distances from our settings to those in practice. However, we believe that it is far more important to build solid understandings, even having distances with practice, than to draw conclusions or make claims that are closer to practice but questionable due to insufficient control over data and evaluations.

## Acknowledgement

The authors would like to thank colleagues from the OSU NLP group and CMU NeuLab for their thoughtful comments. This research was supported in part by NSF CAREER #1942980, ARL W911NF2220144, NSF OAC 2112606, and Ohio Supercomputer Center [5]. The views and conclusions contained herein are those of the authors and should not be interpreted as representing the official policies, either expressed or implied, of the U.S. government. The U.S. Government is authorized to reproduce and distribute reprints for Government purposes notwithstanding any copyright notice herein.

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

# A Training Details

All implementations were done with PyTorch [45] and Huggingface Transformers [68]. All model training runs are done on NVIDIA A6000 and A100 GPUs and last 96 hours at maximum.

# B Effect of Model Scale

We run the experiments on composition with larger model scales with $|\mathcal{E}| = 2000$ and $\phi \in \{5.4, 9.0, 18.0\}$. The results are shown in Figure 7. Overall, it could be seen that scaling up the model won't qualitatively change the model's generalization behaviors, and the main pattern is that larger models converge in fewer optimization steps, which shares with prior findings [60, 28].

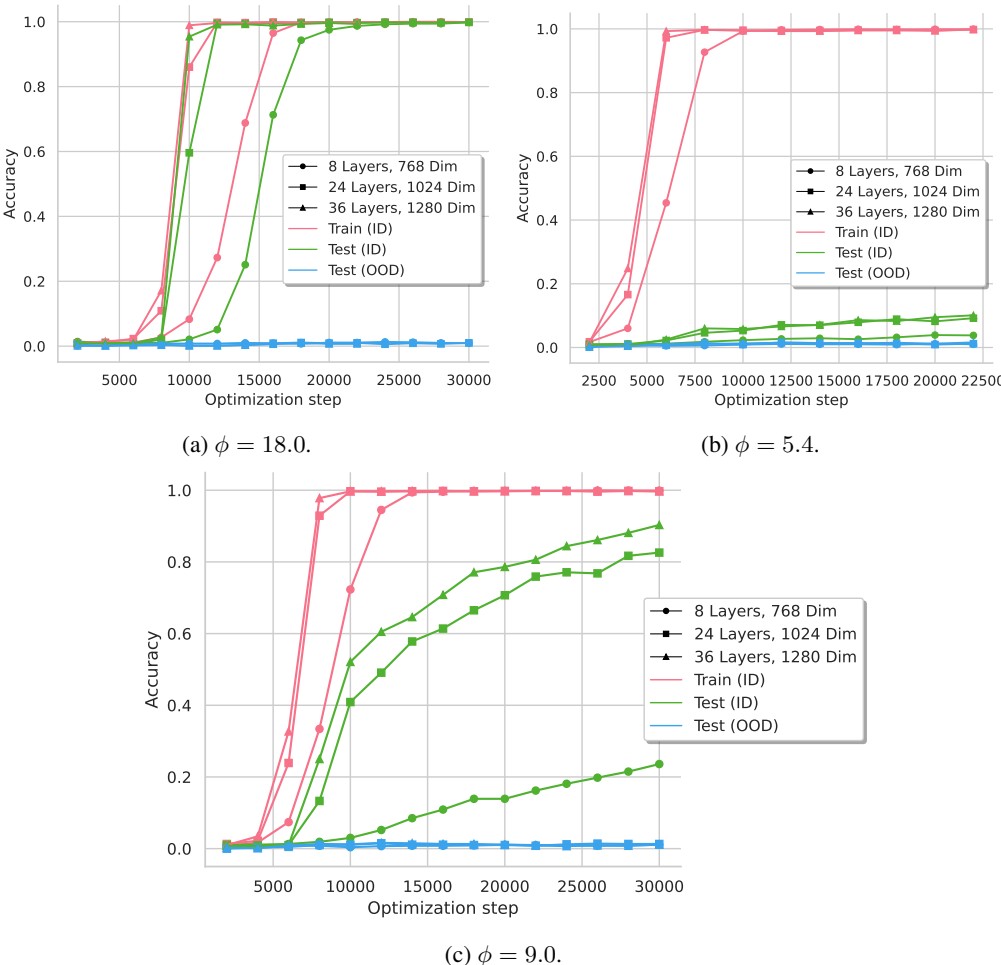

(a) $\phi = 18.0$.

(b) $\phi = 5.4$.

(c) $\phi = 9.0$.

Figure 7: Results for different model scales across different $\phi$. Larger models converge in fewer optimization steps, but have no qualitative changes on the learned behaviors.

# C Effect of Tokenizations

In the experiments in our main content, tokenization is done by having a unique token for each entity.[12] This is different from how real-world entities are typically tokenized—in practice, entities are usually multi-token, and different entities could share tokens at different positions. We investigate

---

[12]Preliminary experiments show that tokenization of the relations does not exhibit notable impacts, which is expected since relations are always explicitly given.

the effect of tokenizations on the composition task by having two tokens for each entity (resembling the first name and last name of a person) in the setting with $|\mathcal{E}| = 2000$ and $\phi = 12.6$. We generate a set of unique first names and a set of unique last names with equal size from which the two tokens for each entity are randomly chosen (we make sure each entity gets a unique ordered pair of tokens). We define *token multiplicity* to be the number of entities that share the same first/last name. For example, when the size of the set of first/last names is 50, the token multiplicity would be $2000/50 = 40$.

Figure 8(a) shows the ID test results, where the training and OOD results are the same from earlier (training performance saturates quickly, OOD result remains zero). It can be seen that a larger token multiplicity would delay the generalization, which is expected to a certain degree since the scale of the model is effectively smaller due to having fewer tokens in the vocabulary. Nevertheless, ID generalization always happens. We also run linear probing on $S[5, r_1]$ throughout training to predict the second token of the bridge entity $b$ in the setting with token multiplicity $40$,[13] where the results are shown in Figure 8(b). It can be seen that the second token of $b$ can be perfectly decoded from $S[5, r_1]$ after grokking, and the decodability improves throughout grokking. This suggests that for the multi-token case, the model is additionally storing the second token of $b$ into $S[5, r_1]$ throughout grokking, which may be another factor that further delays the speed of grokking. These results also share with recent findings that in many cases, tokens beyond the immediate next token are linearly encoded in the hidden states [69, 44, 2, 4].

In summary, different tokenizations affect the results in rather expected ways, and do not influence our main findings and conclusions.

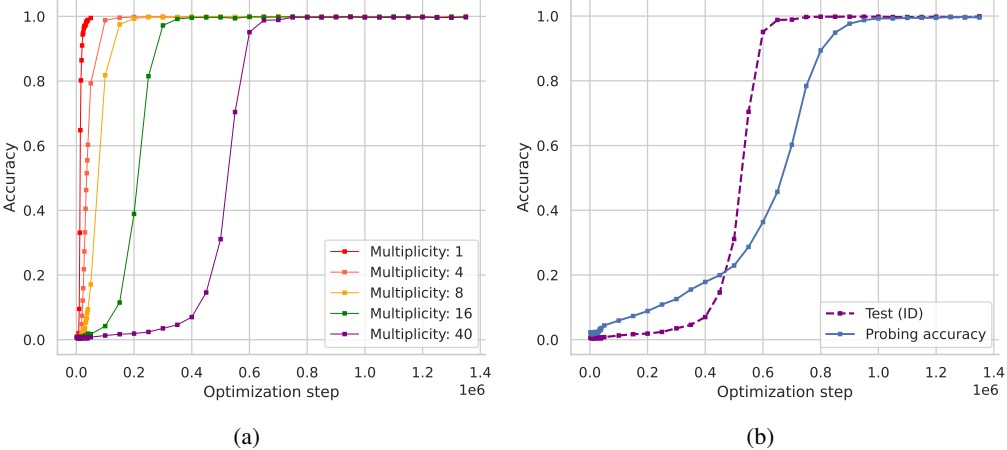

(a)  (b)

Figure 8: (a) ID generalization across different token multiplicity. (b) Probing accuracy on the second token of $b$ at $S[5, r_1]$.

# D  More Details on Circuit Analysis

## D.1  Composition

We run causal tracing on hidden states in layer 1-7 and every position, where the target is the final prediction state $S[8, r_2]$. The changes in the strengths are monotone and smooth, and we show in Figure 9 the strengths for the model checkpoint at the start and end of grokking, and also their difference (same as Figure 4(b)). We also find that after grokking, the state $S[5, r_2]$ (which encodes $r_2$) is not affected by perturbing the input $h$ or $r_1$.

**Deriving the generalizing circuit**. Starting from the 9 states in layers $0, 5, 8$ we can directly eliminate $S[8, h]$ and $S[8, r_1]$ since they have no computational paths connecting to $S[8, r_2]$. $S[5, h]$ can be eliminated as could be seen by Figure 9(c). The connections from $S[0, h]$ and $S[0, r_1]$ to $S[5, r_2]$ could be eliminated as mentioned earlier.

---

[13]Recall that $S[5, r_1]$ is the state that encodes the bridge entity in the generalizing circuit (Figure 4(a))

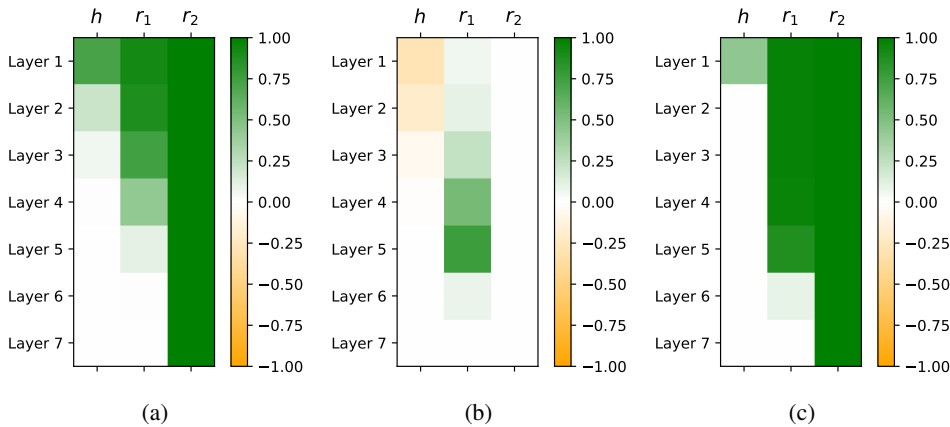

(a)             (b)             (c)

Figure 9: Causal strengths on composition task, with the final prediction $S[8, r_2]$ as the target. (a) Start of grokking. (b) Change during grokking. (c) End of grokking.

## D.2   Comparison

Figure 10 includes the causal tracing results where the target is the prediction state $S[8, e_2]$, and Figure 11 includes the results with the target state $S[5, e_2]$. It can be seen that after grokking, $S[5, e_2]$ does not depend on $e_1$, which gives the generalization circuit in Figure 5(a).

Figure 12 shows the rank (via logit lens) of the three relations $\{a_<, a_=, a_>\}$ in the label space at state $S[7, a]$, where we use Recall@3 as the measure. It can be seen that $S[7, a]$ encodes the label space throughout grokking.

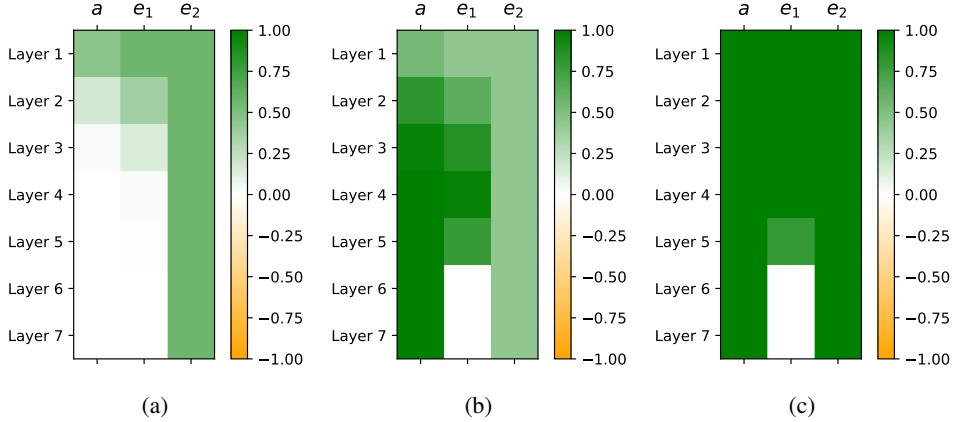

(a)             (b)             (c)

Figure 10: Causal strengths on comparison task, with the final prediction $S[8, e_2]$ as the target. (a) Start of grokking. (b) Change during grokking. (c) End of grokking.

# E   Additional Results

## E.1   Weight decay

Figure 13 shows the ID generalization performance when varying the degree of weight decay ($|\mathcal{E}| = 2000$ and $\phi = 9.0$). It can be seen that a larger weight decay can improve the speed of grokking, and vice versa.

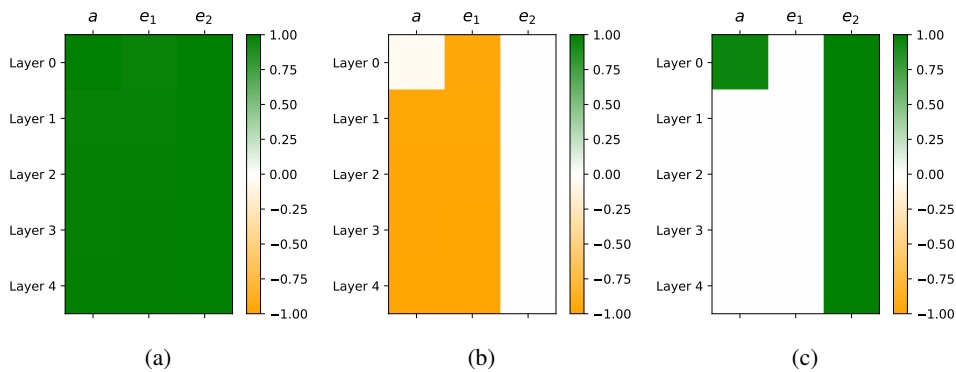

Figure 11: Causal strengths on comparison task, with $S[5, e_2]$ as the target. (a) Start of grokking. (b) Change during grokking. (c) End of grokking.

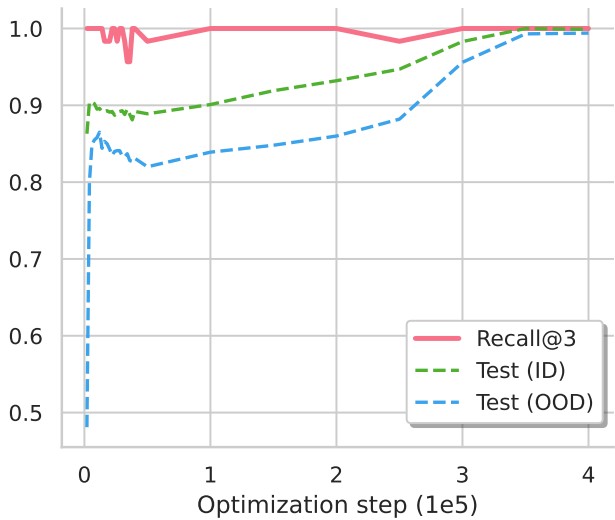

Figure 12: For the comparison task, $S[7, a]$ encodes the label space throughout grokking.

### E.2 Transformer with parameter sharing

We share the parameters of the first $4$ layers and the last $4$ layers, similar as in Universal Transformer [7]. This would allow the model to share the knowledge in the upper and lower layers. The results on the setting with $|\mathcal{E}| = 2000$ and $\phi = 12.6$ are shown in Figure 14. It could be seen that the parameter-sharing scheme can unlock OOD generalization, even though it is gained much more slowly than ID generalization during grokking.

### E.3 Comparison task across different inferred/atomic ratio

Figure 15 includes the result for $\phi \in \{3.6, 7.2, 9.0, 12.6\}$ for the comparison task. It can be seen that a higher ratio $\phi$ would give a higher generalization speed, consistent with the results in the composition task.

## F Complex Reasoning Task

For the complex reasoning task, for each attribute, we include 3% random facts from the comparisons between ID and OOD entities and the comparisons between ID and ID entities. In total, the training set contains 18K ID atomic facts, 437K (ID, ID) comparisons, and 108K (ID, OOD) comparisons, altogether 563K facts (28K on average for each attribute). For translating the facts into natural

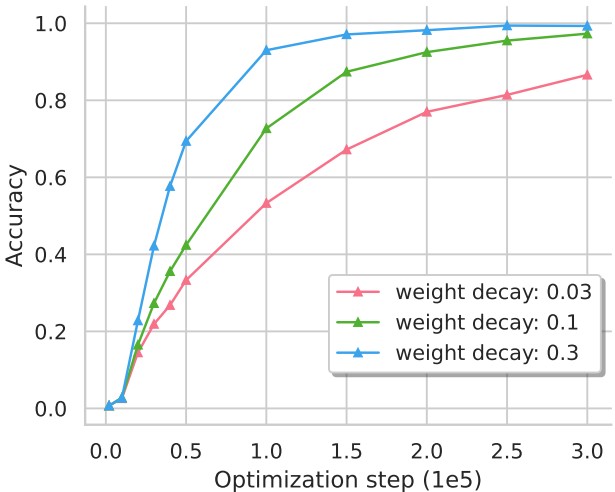

Figure 13: Effect of weight decay. A larger weight decay can improve the speed of grokking, and vice versa.

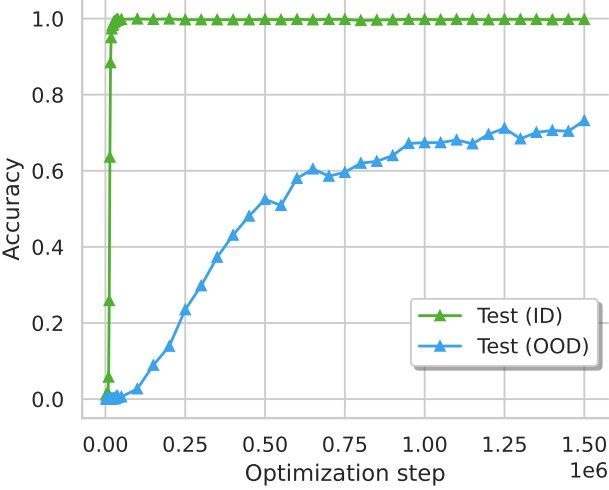

Figure 14: OOD accuracy of the shared-parameter transformer model.

language for testing LLMs with non-parametric memory, we always use the attribute $age$[14] (we find that the choice of attribute does not exhibit notable impact) and the templates "The age of {entity} is {attribute value}." and "{entity 1} is {younger than/older than/in the same age as} {entity 2}." for atomic facts and comparisons. The entities are mapped to distinct random names generated by a random generator.[15] We also try different mappings (e.g., unique IDs) and templates, and find the results to be consistent. All (retrieved) facts are randomly permuted and concatenated before being loaded into the LLM context.

The train/test accuracy, and also the accuracy of inferring the attribute values of the query entities (which we test using the same format as the atomic facts in training) are included in Figure 16. It could be seen that, during grokking, the model gradually locates the ground truth attribute values of the query entities (note that the model is not explicitly encouraged or trained to do this), allowing the model to solve the problem efficiently with near-perfect accuracy.

---

[14]Recall that we test each attribute separately by grouping the facts/queries.
[15]https://pypi.org/project/names/

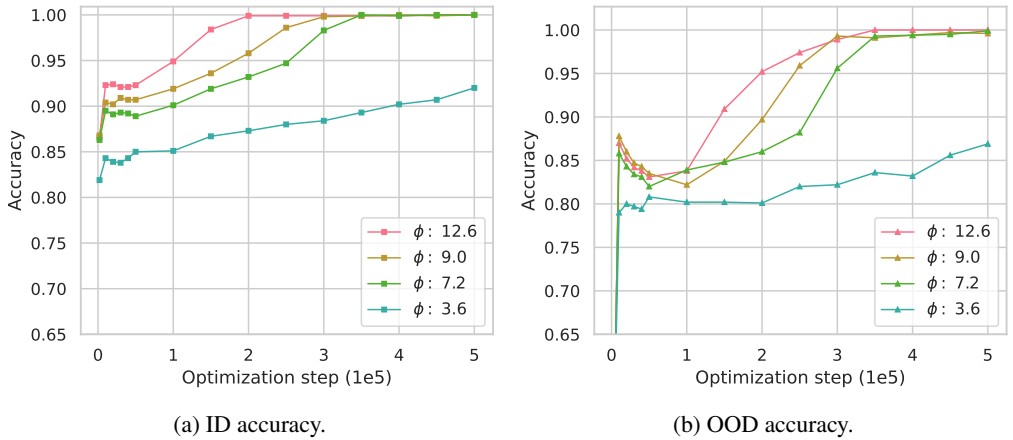

(a) ID accuracy.

(b) OOD accuracy.

Figure 15: Results for the comparison task across different ratio $\phi$.

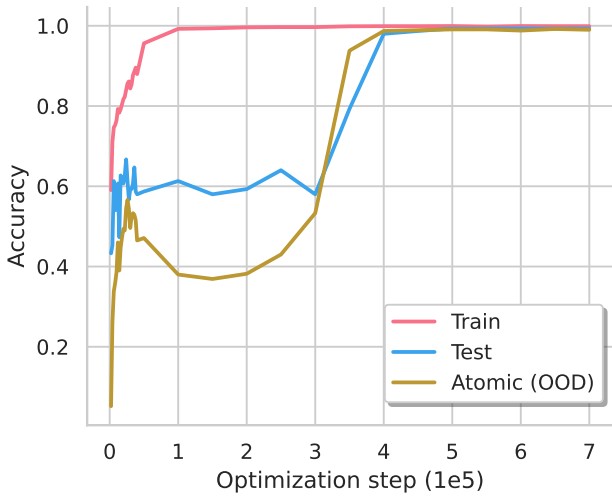

Figure 16: Accuracy on the train and test split, and also the accuracy of inferring the attribute values of the query entities (**Atomic (OOD)**) for the complex reasoning task in §5.

