# OpenReview forum: "Grokking of Implicit Reasoning in Transformers: A Mechanistic Journey to the Edge of Generalization"
_NeurIPS.cc/2024/Conference — NeurIPS 2024 poster_

### Official Review · Reviewer_EZN5 · 2024-07-01

**Soundness:** 3
**Presentation:** 3
**Contribution:** 3
**Rating:** 8
**Confidence:** 4

**Summary:**

Recent work has shown that LLMs are bad at implicit reasoning over parametric knowledge, and this work asks whether that's a fundamental limitation of the transformer architecture or not. Through thoroughly investigating the performance of an 8-layer transformer trained from scratch on 2-hop reasoning tasks, this paper shows that the models can only learn to reason implicitly over parametric knowledge when they learn a "generalising circuit" (as opposed to a "memorising circuit"). They find that in most settings, the model first learns a memorising circuit, and only after grokking (training for many steps after the training set has been fitted), the model learns the generalising circuit. By varying aspects of the training data distribution the authors find that, contrary to what prior work claims, it's not dataset size contributing to grokking happening, but the data distribution (ratio of inferred facts/atomic facts, higher means grokking happens faster). Reasoning about the generalising circuit and when it would be preferred over the memorising circuit, the authors come up with testable predictions for when grokking should happen, why the model does/does not generalise OOD, and what could help it generalise OOD. They empirically show (in the appendix) that these predictions are true in their domain. One of the main insights here is that the generalising circuit is a more efficient representation of data generating distribution, and if you bias a model towards favouring efficient representations with things like weight decay, the model will find it faster.  Another insight is that parameter sharing can help OOD generalisation. The authors finally construct a more difficult, three-hop implicit reasoning task, that requires searching over a larger space to find the "bridges" that allow "hopping" from one fact to another and answer a comparison question. They show that SotA LLMs like GPT-4 cannot do this, but a grokked transformer can do it almost perfectly. The authors take their results to show that it's important for future work on robustness / systematicity of LLM reasoning to consider the power of reasoning over parametric memory with a generalising circuit that more efficiently encodes the task than a memorising circuit.

**Strengths:**

- This paper explains the grokking phenomenon very effectively. The biggest strength of this paper is that they use their mechanistic analyses to make testable predictions about when and why grokking should happen, and then manage to use these predictions to manipulate the phenomenon to show up earlier in training. Additionally, they use their findings to propose improvements to LLM training / architecture that might help it reason more robustly in the future. Of course, it remains to be seen whether this will hold at scale, but the authors convincingly show that these improvements help a smaller single-task transformer to learn to generalise OOD and robustly reason, and that reasoning robustly (in this particular rule-based way) doesn't need to be a general limitation of transformers.

- Each figure in this paper is exceptional and supplements understanding of the work very well, making the presentation overall great.

**Weaknesses:**

- The paper does a lot, which is a strength, but also means that some details that are required to understand the paper better seem left out. I will ask questions below about these details.

- Not so much a limitation of this work itself, but of one of the methods used (causal tracing). I'm wondering how well causal tracing would work for problems that require using multiple different hidden states in a distributed way, and whether it's mainly for small transformers on relatively simple / rule-based tasks. It seems unlikely LLMs rely that much on a single hidden layer for predictions.

- It's not entirely clear how/if the findings from this work would transfer to very general models like LLMs. Is it reasonable to expect the same recommendations (on weight decay, parameter sharing, using parametric memory to efficiently store atomic facts and learn to apply rules over them) to hold for models like LLMs, that  currently do one-epoch training and do not overfit, let alone far exceed overfitting, on a specific task.  Intuitively, it seems unlikely that we can get both very general models and "grok" each reasoning problem it needs to be able to do, even if we make the generalising circuit more favorable by parameter sharing or weight decay. I do think the authors somewhat address this point in the limitations, and also do not claim their recommendations should hold for every reasoning problem, just those that are well described by rule-based deduction and inference. Nonetheless, I'd be interested to see some more discussion on this.

**Questions:**

Questions
- Do you think the recommendations to make finding the generalising circuit easier will transfer to LLM training?
- Do you have an explanation for why the model learns a parallel pathway for the comparison task and not the composition task?
- What does it mean for the model to "prepare the label space"? Is it that, according to the unembedding at those hidden states, the top tokens are always those in the label space?
- Why does the memorising circuit need to also store the atomic facts and not just the inferred facts for the composition task?
- Line 85-96 about "what happens during grokking". Why do you say that these results indicate the model is mainly memorising without going through the hop before grokking? From the observations you mention it could still be that b is encoded/retrieved elsewhere in the model no? Or did you find that $b$ doesn't get strongly encoded anywhere before grokking?

Writing suggestions:
- Can you give an example of an actual input and output pair you train the models on in the main text?
- Some details about how you discover the generalising circuit are still missing. How many examples do you use to determine the ratio of alterations that is called the "causal strength"? You say you eliminate / prune certain connections, what does that mean, you leave them out and the accuracy of the remaining model is still high? What does it mean that the lower layers retrieve the first-hop fact? Is it that they all unembed to top-1 (or top-3) tokens of the first-hop fact (h, r1, b)?

**Limitations:**

Addressed in the appendix.

---

> ### Author Rebuttal · Authors · 2024-08-05
>
> Thank you for the informative and constructive comments! We will incorporate them to improve our work in the revised version.
>
> **[Causal tracing - W2]** First of all, as you mentioned, we believe this is not a weakness of our work and more of a general question about causal tracing. To explain here, causal tracing (also called activation patching) is a general method and there are a lot of different instantiations based on different views of the causal graph of transformers in different granularity levels. For example, one can do casual tracing at finer granularity (e.g., attention heads, attention QKV matrices, and MLP neurons) or coarser granularity (e.g., groups of hidden states across different layers and token positions). The case you mentioned could definitely be discovered through the method in principle, though this may be quite challenging (due to a large search space). Causal tracing also does work in larger-scale pretrained models and less rigid tasks (e.g., [1] and also many others).
>
> **[Connecting to general models - W3, Q1]** Yes we do admit that our study overall has some distance in connecting to large general models trained on broad corpora and tasks (Appendix H). Nevertheless, currently we have rather little understanding of even the simplest tasks/settings/models, and we believe our work lays a good foundation for subsequent work to study more general and realistic settings. A good concrete direction related to “generality” here would be to study how things behave when we do multi-task training , where the memorizing/generalizing circuits from different tasks are mixed together. For the problem you mentioned (hard to make a general model “grok” on every task in practice), there are also many interesting directions such as how to accelerate grokking (e.g., [2]) and how to better prepare the training data. For example, if the model architecture is good for a rule where it could systematically generalize, then we don’t really need to gather a lot of data for the rule - instead, we just need to curate a small but high-quality subset of data (presumably with high inferred/atomic ratios), and train the model extensively on this data till it gets the rule well (which shouldn’t cost a lot of compute). Then the model should be able to apply the rules when it is learning about other domains. Overall, there are many interesting follow-ups of our work on moving to practical settings, and we will add more discussions in the revised version.
>
> **[Parallel circuit - Q2]** This is determined by the nature of the tasks and the transformer architecture (which processes tokens in parallel). For the comparison task, the two underlying atomic facts have no order dependence on each other and both the query entities are present in the context tokens, while for the composition task, the second-hop fact depends on the (object entity of) first-hop fact, which is why the model can only learn a “serial” circuit.
>
> **[Label space - Q3]** Yes. We included these in Appendix D.2 (lines 575-577). We will add them to the main content in the revised version.
>
> **[Memorizing circuit for composition - Q4,5]** The atomic facts always need to be stored in the weights, and here since the model is very likely directly memorizing the inferred facts without utilizing the stored atomic facts, it separately stores all the inferred facts. To explain further our reasoning around the memorizing circuit: first, the state S[5,r1] encodes the bridge entity throughout grokking, however, its causal connection with the prediction state is very low (Figure 4c, Appendix D.1) in the beginning (which grew significantly during grokking). Indeed we cannot rule out the possibility that the model is somehow using the bridge entity information in some other hidden way elsewhere before grokking, but this is unlikely since then the model has to “force itself” not to use the state S[5,r1] which we know already encodes the bridge entity very well. One future direction to check this further is to understand in finer granularity the circuits and their evolvement throughout grokking, which may also help understand deeper the findings here and inspire theoretical investigations.
>
>
> **[Writing suggestions]** Thank you for the suggestions;  we will incorporate them to improve the writing in the revised version. For the ratio, we mean the percentage of the 300 random examples where performing causal intervention changes the target prediction. More details regarding the circuit discovery are included in Appendix D. Yes, by pruning we mean intervening (via the perturbed run), and we prune an edge/node if pruning it negligibly affects the accuracy. By “lower layers retrieve the first-hop fact”, we mean the “left-right” component of the circuit (Figure 4a) where the input states of h and r connect to S[5, r1] which encodes the bridge (via logit lens).
>
> ====
>
> References
> - [1] Lieberum et al. Does Circuit Analysis Interpretability Scale? Evidence from Multiple Choice Capabilities in Chinchilla. arXiv-23.
> - [2] Lee et al. Grokfast: Accelerated Grokking by Amplifying Slow Gradients. arXiv-24.

---

> > ### Comment · Reviewer_EZN5 · 2024-08-10
> >
> > Thanks for answering my questions. I am going to raise to an 8 and will strongly recommend accepting this paper. It very neatly shows how the grokking phenomenon works.

---

### Official Review · Reviewer_smMo · 2024-07-05

**Soundness:** 3
**Presentation:** 3
**Contribution:** 3
**Rating:** 7
**Confidence:** 4

**Summary:**

The work mainly focuses on the systematic generalization (specificially, in the paper, two implicit reasoning types: composition and comparison of facts) of the grokked (i.e., training far beyond overfitting so that the model can finally learn some specific generalization skills and achieve high testing performance) transformer-based language models.

The paper presents empirical findings that the trained transformers: (1) for composition reasoning, grokking is observed in in-distribution generalization testing but not in out-of-distribution generalization testing; and (2) for comparison reasoning, have grokking is observed in both in-distribution and out-of-distribution generalization testing.

The paper also interprets the phenomenon of grokking by investigating the inner working patterns by correlating generalization performance with different reasoning circuits implicitly performed in the models.

Besides, the paper designs a synthetic complex reasoning task, where state-of-the-art LLMs (GPT-4-Turbo and Gemini-1.5-Pro) achieve low performance even with advanced prompting techniques and retrieval augmentation while grokked small-size transformer achieve good performance, stressing the power and the potential of parametric memory inside language models.

**Strengths:**

1. The topics of this paper, the grokking and systematic generalization of transformer-based language models, are very important, timely and of interest to the NeurIPS community. Besides, the paper is overall well-written with fluent presentation and clear logic structure, thus being easy-to-follow.

2. The study presented in the paper is comprehensive: from observing generalization performance to investigating the internal reasoning circuits. The derived reasoning circuits in the grokked transformers can explain the different generalization behaviours in different settings (for both in-distribution testing and out-of-distribution testing; for both the composition reasoning task and the comparison reasoning task) very well.

3. *From the perspective of implicit reasoning in LMs*, I appreciate that the authors correlate the implicit reasoning (for both composition reasoning and comparison reasoning) with the grokking, demonstrating that LLMs have potential to perform implicit step-by-step reasoning rather than simple memorization, which is a good complement to existing works [1, 2, 3]

4. The results for complex reasoning are quite impressive to me, demonstrating that well-trained ("grokked") transformers with parametric knowledge have potential to largely surpass the state-of-the-art large models with techniques like chain-of-thought prompting and retrieval augmentation.

[1]: Towards a Mechanistic Interpretation of Multi-Step Reasoning Capabilities of Language Models. https://arxiv.org/abs/2310.14491 .

[2]: Do Large Language Models Latently Perform Multi-hop Reasoning? https://arxiv.org/abs/2402.16837 .

[3]: Understanding and Patching Compositional Reasoning in LLMs, https://arxiv.org/abs/2402.14328 .

**Weaknesses:**

My major concerns are two-fold:

1. Though the mechansitic interpretation results to the grokking of implicit reasoning patterns in this paper are beatiful and persuasive to me, I think its inspiration to analyzing real LLMs pre-trained on a large corpus can be quite limited, given the settings of this work (i.e., structured synthetic data, single task, random initalized GPT-2 transformer training from scratch.), as the authors discuss in the Appendix H.

2. *From the perspective of grokking*, there have existing works (to name a few, [1,2]) discussing the mechanism of the grokking phenomenon: memorization versus generalization, the transition and competition of the implemented circuits inside models (most of which focused on different tasks, e.g., algorithmic tasks). I think though this work specifically focuses on knowledge reasoning, its conclusions and insights partly share with previous works so that they may not that novel to me.

[1] PROGRESS MEASURES FOR GROKKING VIA MECHANISTIC INTERPRETABILITY, https://arxiv.org/abs/2301.05217 .

[2] Do Machine Learning Models Memorize or Generalize? https://pair.withgoogle.com/explorables/grokking/ .

**Questions:**

(1) For my major questions and concerns, please refer to the "Weaknesses" part.

(2) minor question: In figure 4 (a), how do you demonstrate it is the bridge entity (**b**) encoded in the hidden state of (Layer 5, r1) position that causally affect the final prediction (**t**). In the causal tracing procedure described in the paper, I can only infer that the whole hidden state has causal effect on the final prediction (what if something else in the hidden state plays the important role?)

(3) minor question: I am wondering what if we generalize the composition reasoning reasoning from two-hop to three-hop and even more hops? Do the grokking phenomenon still emerges? It seems that the complexity of circuits implemented by transformers is bounded by their layer.

(4) minor question: I find detailed description, settings and results for Table 4 (e.g., CoT prompt examples, specific error cases) are missed in the paper. I am wondering why the performances with chain-of-thought prompting are even worse. Could the authors please provide more detailed results and insights to this point?

(5) A minor suggestion: this work [1] also discusses the mechanism of compositional factual reasoning in the LLMs, suggesting LLMs can implicitly perform mutli-hop reasoning and the reasoning errors might stem from improperly leveraging the bridge reasoning results (which are also in align with the observations presented in this work). Hence it is good (not necessary) to also discuss this work in the Related Work section.

To conclude, though I have concerns on the generality (to widely-used LLMs) and novelty (from the perspective of grokking) of the paper, I appreciate its organic combination of grokking, implicit reasoning and mechanistic circuit analysis. I think the work will clearly have its impact on the LLMs' implicit reasoning (with the parametric knowledge) field and hence lean towards accepting it.

[1]: Understanding and Patching Compositional Reasoning in LLMs, https://arxiv.org/abs/2402.14328 .

**Limitations:**

Yes, the authors do include a section (Appendix H) to discuss the main limitations of the work. I would also suggest the authors discuss the limitations of Logit Lens [1,2] (used for analyzing the internal reasoning circuits) in this section.

[1] Logit Lens: https://www.lesswrong.com/posts/AcKRB8wDpdaN6v6ru/interpreting-gpt-the-logit-lens .

[2] Eliciting Latent Predictions from Transformers with the Tuned Lens, https://arxiv.org/abs/2303.08112 .

---

> ### Author Rebuttal · Authors · 2024-08-05
>
> Thank you for the informative and constructive comments! We will incorporate them to improve our work in the revised version.
>
> **[Abstract nature and connection to practice - W1]** Yes, we do admit that one limitation of our work is its synthetic and abstract nature (Appendix H). Still, we believe that our work is a firm initial step that future work could build on and gradually move to more realistic settings.
>
> **[Insights on grokking - W2]** While we indeed borrow ideas and existing developments in grokking to understand and explain our findings, we note that we also bring novel aspects and potential corrections of prior explanations of grokking through our controlled study. Specifically, through controlled experiments (lines 122-143), we find that existing explanations based on “critical data size” are potentially flawed or not general enough, and it could instead be “critical data distribution” that is really the crucial factor.
>
> **[Limitation of logit lens - Q2, Limitations]** This is one limitation of logit lens, or more generally, all interpretability methods that are based on “lossy” projections (including further improvements of it such as tuned lens as cited, etc.). This is still an open problem for interpretability research to the best of our knowledge, and we currently don’t have a good way to resolve this. Some potential directions may be to do further interventions on the “remaining components” (e.g., projection of the state to a subspace orthogonal to the bridge entity’s embedding) but these could again require further justifications. Nevertheless, despite the possibility of alternatives, we believe that it is reasonable to conceptualize the state as the bridge entity in our case here given the strong correlation, and will discuss these limitations in the revised version.
>
> **[Composition with more hops - Q3]** This is one interesting direction for follow-up studies; here based on our findings, we believe that grokking should still happen generally (depending on the inferred/atomic ratio) and the vanilla transformer would certainly still suffer in systematic generalization. Indeed it would be difficult for a vanilla transformer to resolve a large number of hops given their bounded computations; one related interesting direction would be to investigate whether cross-layer memory-sharing models such as universal transformers (we also briefly investigated this in Appendix E.2) could solve this and maybe even generalize to more steps than those seen in training (when the layers are allowed to be executed with unbounded steps).
>
> **[Details about the results and settings in Table 4 -  Q4]** Indeed we omitted a lot of details here (we also put some details in Appendix F) in the paper due to space limit, and we will add them in the revised version. For the CoT prompt, we are using the zero-shot instruction which encourages the model to “think step by step”. Regarding why adding CoT makes the performance worse, first of all, the model’s performance without CoT is still pretty close to random guesses, so a summary would be both settings fail badly. Now specific to the performance drop, we found that with CoT, 70.7% of Gemini’s responses ended up saying that the answer cannot be decided (which we treat as wrong since the answer can be decided). The ratio drops a bit to 58.7% when augmenting with retrieval. One typical example model response is included in Table 1 in the added PDF. Intuitively this does make a lot of sense because, when the model is instructed to verbalize the reasoning, it is harder for it to “guess” the answer because the generated rationales are present in the context and the model would have more explicit clues that the logic doesn’t really work out.
>
> **[Related work - Q5]** Thank you for referring to the related work here. We will discuss it in the revised version.

---

> > ### Comment · Reviewer_smMo · 2024-08-10
> > **Thanks for the authors' response**
> >
> > Dear authors,
> >
> > Thanks for the detailed response, which resolve part of my conerns. I maintain my initial score to this paper.
> >
> > Reviewer smMo.

---

### Official Review · Reviewer_Xiqz · 2024-07-10

**Soundness:** 2
**Presentation:** 3
**Contribution:** 2
**Rating:** 6
**Confidence:** 4

**Summary:**

This paper investigates whether Transformer models can learn implicit reasoning through the phenomenon of "grokking" focusing on two types of reasoning: composition and comparison. Also, the paper reveals the mechanisms behind grokking and the differences in systematic generalization across different reasoning tasks through internal model analysis.

**Strengths:**

1.	The paper explores the relatively unexplored phenomenon of "grokking" in the context of Transformer models, offering new insights into how extended training can lead to implicit reasoning capabilities.
2.	By focusing on both composition and comparison reasoning tasks, the study provides a thorough examination of the different ways Transformers can generalize based on the circuit mechanism.
3.	The paper analyzes in-distribution (ID) and out-of-distribution (OOD) generalization for composition and comparison reasoning tasks. It explains why the composition task cannot generalize systematically while the comparison task can, by examining circuit mechanisms.
4.	The paper trained a grokked transformer to finish the more complex reasoning task which is far beyond the SOTA like GPT-4-Turbo and Gemini-1.5-Pro.

**Weaknesses:**

1.	The paper does not provide how to tokenize the whole sentence in (1)(2) and how to evaluate the loss function.
2.	The beginning epoch of the abscissa in Fig.1 needs clarification. Why the first point in Fig1b Train accuracy is about 0.9?
3.	Since the accuracy is a discontinuous metric, it may sometimes mislead for indicating grokking. The loss for each training needs to show.
4.	The interval over which the change in causal strengths in Figure 4b is calculated is not specified. Additionally, it is unclear how the change is defined.
5.	The term 'same type' in line 172 needs clarification. Does it refer to data within the same entity set or add noise?
6.	How to calculate the ratio in line 175 needs clarification.
7.	Fig.4b,4c only show the correlation between the causal strengths and the grokking. There is no evidence indicating that the formation of the structure causes the grokking. In other words, this correlation is shown in (Reddy, 2023).
8.	The paragraph titled 'Why does grokking happen?'  in line 197 is inappropriate. It only describes the correlation between grokking and the formation of the circuit, explaining what happens during grokking but not why it occurs. The main question that needs to be addressed is why transformers tend to form this circuit when the training accuracy is high (the training loss is low).
9.	In the circuit of Fig5a, the analysis through the logit lens only shows the output at this layer, but how can the transformer maintain the output to layer 5 is not shown.

**Questions:**

1.	In line 203, what does 'stores... in the weights' mean? Does it imply that during the memory phase, the logit lens is not 'b'?
2.	Is the grokked transformer a newly trained model or is it based on the model described in the previous sections? If it is a newly trained model, how was it trained?
3.	What will happen when we input out-of-distribution data in Figure 4a? Will the logit lens at S[5, r1] not show 'b'?
4.	In my opinion, the failure of the OOD in the composition is due to the unseen embedding of OOD data during the training.

**Limitations:**

See weaknesses.

---

> ### Author Rebuttal · Authors · 2024-08-05
>
> Thank you for the informative and constructive comments. We will incorporate them to improve our work in the revised version.
>
> **[Tokenization and loss - W1]** If the “(1)(2)” here means the rules in Equations (1) and (2), these rules are latent and the model only sees the atomic and inferred facts (deduced from the atomic facts via the rules). This is explicitly mentioned in Section 2. We also stated how we tokenize the facts in the main experiments (lines 102-104, 246-248) and have some further investigations on tokenization (Appendix C). For the loss, we use the standard cross-entropy loss as in normal language modeling. We will add these details in the revised version.
>
> **[Initial point in Figure 1 - W2]** The initial point here is for 2K optimization steps. For reasons why the comparison task has a high performance in the beginning: the comparison task has a significantly smaller effective label space (3 comparative relations for the specific attribute) compared to the composition task (all entities), and also the model could more easily guess the answer correctly, e.g., if an entity’s attribute usually appears to be larger than others’, when the model is asked to compare it with someone else, it could just guess that this entity has the larger attribute without really comparing them.
>
> **[Metric used - W3]** Thank you for the suggestion. We plot the loss curves in Figure 1 of the added PDF under the same setting as Figure 1 in our paper. It could be seen that the loss curves have the same trends (upside down since small means better for loss) as the accuracy curves. This clears out the concern here.
>
> **[Details in causal tracing - W4,5,6]**
> - The interval is taken to start from the beginning of grokking (when training performance saturates) to the end of grokking (when test performance converges). The change is defined in the natural way, which is the causal strengths at the end minus the strengths at the beginning of grokking.
> - ‘Type’ here means entity/relation. For example, we are not perturbing an entity into a relation.
> - The ratio is calculated as the percentage of the 300 random examples where the causal intervention changes the prediction.
>
> We will make these details more explicit in the revised version.
>
> **[Explanation of grokking - W7,8]** We are suggesting a plausible explanation of what we observe here through the lens of circuit efficiency, with concrete evidence (e.g., comparison of the amount of facts different circuits store) and further experimental consolidations (Appendix E.1). A more fine-grained and rigorous answer to why grokking happens would require further efforts such as detailed reverse-engineering of the weights and analyzing the training dynamics. These are interesting follow-up directions based on our work, but we believe they are also beyond the scope and amount of effort of this paper. We are not sure which paper you are referring to by "(Reddy, 2023)"; it would be great if you could further clarify this.
>
> **[Detailed mechanisms within the circuit - W9]** Related to the last point, while a more detailed and lower-level understanding of how the model does the computations within the circuits is interesting to have, these won’t really affect our conclusions as we are not doing fine-grained (theoretical) analysis in this work. We believe that our current work already presents significant efforts and provides highly insightful findings and analyses, which could serve as a solid foundation for subsequent study at more detailed levels.
>
> **[Q1]** The logits lens at S[5, r1] is ‘b’ from the beginning, however, the causal strength between S[5, r1] and the target state is very low (lines 189-190, Appendix D) which suggests the model is not traversing the bridge and directly memorizing the inferred facts.
>
> **[Q2]** It is a model trained (from scratch) on the given facts, not directly taken from previous experiments. We will make this clearer in the revised version.
>
> **[Q3]** We did compute the logit lens results for the OOD setting, and found that S[5, r1] encodes the bridge entity (with MRR 0.98), which is a strong indication for our subsequent explanation in lines 216-217. We will include these results and expand the discussion in the revised version.
>
> **[Q4]** The OOD facts actually share the same set of entities and relations as the ID facts (see Section 3.1). Moreover, we did experiments on an alternative architecture with cross-layer parameter-sharing and found that it could achieve impressive OOD generalization in composition (lines 226-228, Appendix E.2), which implies that embeddings are not the major issue and also consolidates our explanations in the paper.

---

> > ### Comment · Reviewer_Xiqz · 2024-08-10
> >
> > I refer to the paper 'The mechanistic basis of data dependence and abrupt learning in an in-context classification task' (Reddy 2023). Several hidden progress measures are implemented to show the grokking of ICL (In context learning) in that paper. If there is any relationship with your results?
> >
> > And I have no further inquiries. Based on the responses provided, I will be revising my evaluation favorably.

---

> > > ### Author Response · Authors · 2024-08-14
> > >
> > > Thank you very much for your support of our efforts. Also many thanks for clarifying the reference. It is a very insightful analysis for in-context learning, which is very different from our focus (i.e., implicit reasoning), though. The progress measures defined in the paper are inspiring. It'd be very interesting follow-up work to investigate similar quantitative measures and visualize them during the grokking process of our studied tasks. We will discuss this work and point out related future directions in the revised version. Thanks agains for your efforts reviewing our work and for all the great comments!

---

### Official Review · Reviewer_f5kN · 2024-07-12

**Soundness:** 3
**Presentation:** 3
**Contribution:** 2
**Rating:** 5
**Confidence:** 3

**Summary:**

This paper explains the reasoning ability in LMs is acquired through grokking,  which requires extended training beyond overfitting. Through analytical experiments, the authors explore the mechanisms behind grokking, the formation of generalizing circuits, and the impact of systematicity in the configuration of these circuits.

**Strengths:**

1. Novel Insight. critical data distribution decides the characteristics of grokking
2. Sufficient Mechanistic Analysis. The detailed mechanistic analysis of the internal workings of transformers during grokking offers valuable insights into the formation of generalizing circuits and their efficiency.
3. Clear Presentation. The study is well-structured, with clear explanations of the experimental setup

**Weaknesses:**

1. Limited Scope. The focus on only two types of reasoning (composition and comparison) may limit the generalizability of the findings to other reasoning tasks.
2. Dependence on Synthetic Data. The experiments primarily use synthetic data, which may not fully capture the complexities and nuances of real-world data.

**Questions:**

1. Can you provide the detail of your data sizes ? i can find the statics and samples of your data.
2. any other reasoning task can confirm your assumptions ?

**Limitations:**

See in Questions.

---

> ### Author Rebuttal · Authors · 2024-08-05
>
> Thank you for the constructive comments! We will incorporate them to improve our work in the revised version.
>
> **[Limited scope and dependence on synthetic data]** We do admit these limitations in the paper (Appendix H), however, we believe that despite these, we are taking the initial steps toward studying the problem by laying out a clear and well-defined formulation, and also conducting rather deep investigations on the tasks we study. This could serve as a solid foundation for future work to build on and move to more realistic settings.
>
> **[Details of data sizes]** For the task of composition, the number of total atomic facts is 20*|E|, where 95% are utilized as ID facts and the remaining ones are OOD facts. The specific quantities of the facts would then depend on 1) the number of entities |E| and 2) the inferred/atomic ratio, which we vary in our experiments to see their effects on the model. For example, when |E|=2K and ratio=9.0, there will be 38K ID atomic facts, 2K OOD atomic facts, and 342K ID inferred facts that go into the training data. The case of comparison is similar. We will make some of these quantities explicit in the revised version.

---

### Official Review · Reviewer_jEH8 · 2024-07-15

**Soundness:** 1
**Presentation:** 2
**Contribution:** 2
**Rating:** 5
**Confidence:** 4

**Summary:**

This paper explores the ability of transformers to learn two synthetic tasks when trained from scratch. The tasks consist of rigidly structured data and effectively represent: 1. following a path of length 2 in a graph (the "composition" task), and 2. looking up + comparing two items in a dataset ("comparison").

The investigations find that the transformers only manage to generalise the tasks after "grokking", a phenomenon in which test performance improves long after train performance has saturated, and moreover that the composition task does not generalise as strongly as the comparison task.

Causal tracing is done on the transformers to gain an intuition of how they solve the tasks, i.e. revealing their internal "circuits". For the composition task, it is found that the network looks up individual edges of the graph at two different stages, whereas for the comparison task all information can be stored in a single subsection for the solution to work, providing some intuition on their different generalisation potential.

Some manipulations of the datasets are done to see how this influences training, with conclusions drawn on dataset size and on the ratio of types of data. The ability of two commercially available LLMs to perform the task in a zero shot setting, when given the same data in-context, is also evaluated.

**Strengths:**

Through causal tracing, the manner in which a transformer may solve one of two types of tasks - amounting to chained retrieval ("composition") or parallel retrieval ("comparison") - is found. As may be expected, in chained retrieval the transformer must access data at two different points in computation, while in parallel retrieval it may retrieve all data from a single 'area' (part of its weights) - making parallel retrieval more easy to generalise. While this is not surprising, it is nice to see explicitly, and shows an interesting new case where weight sharing or universal transformers may (i note may, as this is not investigated) have an advantage over vanilla transformers (i previously thought the benefit of universal transformers were restricted to their adjustable computation depth, so this is nice). The conclusion on the more complicated composition mechanism is supported also by the investigations of generalisation ability (in and out of distribution test performances) of transformers on the two tasks.

**Weaknesses:**

1. Overclaiming: Framing should be adjusted to more clearly reflect the work done.
- Title overblown for an investigation of two rigidly structured synthetic tasks. I highlight: the structure is so rigid that all input samples have one of two short fixed lengths: 3 tokens for single datapoints, and 4 for demonstrations of composition/comparison.
- In general I feel people (not just in this paper) are too quick to make claims on 'reasoning' from experiments on straightforward formal tasks (which are valid and interesting in themselves without such aggrandizations, not to mention neater to discuss without them). In the case of this paper, the framing is around reasoning, but in practice it investigates the ability to learn instances of two highly structured synthetic tasks (instances of: the tasks are not learned in a general sense, but rather, with respect to specific datasets).
- The suggestions for how transformers/training should be improved are either poorly supported by the experiments or not investigated beyond speculation: see comments 12, 13, 14, and 34. Correcting these analyses and doing the investigations to support the various suggestions would significantly improve the paper.
- Unfair comparison to parametric memory, see comment 35
2. Poor analysis of results: in particular, see comments 12, 13, 14 regarding manipulating the ratio of inferred data in the dataset, or comment 24 regarding conclusions on internal mechanics of the model.

**Questions:**

1. The (recurring) statement that GPT4-Turbe and Gemini-1.5.Pro are "based on non parametric memory" (lines 16-17, and later 64-65, 280-281, 307) is confusing and potentially misleading - the two obviously also have significant parametric memory. The intention here is more about their ability to reason over *additional* knowledge provided in a non parametric way, i.e., this is about comparing reasoning over in context knowledge vs trained (parametric) knowledge. I would rephrase to be clear on that: this has been a comparison of zero shot in context learning vs task- and dataset- specific pretraining. I imagine there are works already talking about the fact that fine tuning is stronger than in context learning.
2. line 25, English: an implication is not an impact or vice versa, rephrase to align
3. line 28 define systematic generalisation
4. intro style comment: try to reserve italics only for when a new term is being introduced (in which case it should be accompanied by a useful definition). can use very sparingly for emphasis but this is a lot. generally, a lot of terms do fly around here that would benefit from description, but are not accompanied by one - implicit reasoning, systematic generalization, mechanistic analysis, parametric memory, etc.
5. general comment: can we really say that a model has learned to reason (as claimed in this paper) when it does not generalise out of distribution (composition task)?
6. fig 1 last sentence shouldn't be here - caption is for explaining figure, not promoting other parts of paper
7. lines 55-59 too vague by themselves, i get nothing from this part of the intro. subsequently, lines 59-61 not meaningful until after reading entire paper - which is not the goal of an intro! be more concrete.
8. missing: clear numbers on dataset sizes and in particular repetitions while training. when you made phi larger did that make the dataset larger or did it mean more repetition or..? do i understand correctly that the maximum number of possible inferred facts is |E| x 20 x 20 (subject, outgoing relation 1, outgoing relation 2)? if so for |E|=2000 as in line 113, that means 800k inferred facts (and slightly less for the actual trainable inferred facts ie those not set aside as either OOD or even ID test) and 20 x |E|=40k atomic facts. given batch size of 512, thats only enough for <2k optimisation steps .... was there a huge amount of epochs here? (line 120: training goes to 2 million training steps)
9. from things like figure 3 and equation 1, i infer that the input sequences are presented as short structured 3- or 4- (for atomic vs inferred facts) token sequences, but this should have been explicitly stated and presented.
10. regarding the facts graph: in a real set of facts, some relations do not make sense: for example, a number entity should not be the target of a person relation (e.g. 'barack's wife is 1964'). it seems the graphs considered in this work do not reflect such constraints. that's fine for an exploration of graphs, but not great for drawing conclusions on general reasoning skills. again, it is my opinion that this paper should be reframed to reflect its more straightforward graph processing nature. it can of course be accompanied by some discussion on how this may relate to reasoning in more 'general' models, and what insights may be taken from this research for work on such models.
11. lines 122,3: "ID accuracy", "other splits" - be more explicit ("test ID accuracy", "train ID and test OOD") else hard to read
12. figure 2: what happens at "ratio" (caveat: see next comment) greater than 18, i.e., some of the atomic facts not given? what happens at ratio infinity? could be interesting. especially given the fact that it doesn't generalise to the OOD facts, i think its building at least one of the layers of atomic facts from the composed samples.
13. in the experiments framed around ratio between atomic and inferred facts, it is something else that is being directly varied: the percentage of the inferred facts given in the dataset. this is probably also why the investigation was capped at the highly specific ratio 18, instead of continuing to increase the ratio and find a sweet spot: ratio 18 is the maximum for this dataset when doing this variation (effectively it means, take "90%" of the data, i.e., include all facts that have not been reserved for the test/validation sets). Reporting it in terms of percentage of inferred facts included would be a much clearer and more generalisable framing - the magic number 18 is obviously specific to this dataset. To actually investigate ratio, one should hold fixed the percentage of inferred and of atomic facts being included in the data, and then repeat them different amounts of times in the dataset to change the ratio between them (recall there is a huge amount of repetition happening in training here anyway, see comment 8). the conclusion of line 125-126 now becomes the much clearer "providing maximum samples of inferred data speeds up generalisation".
14. the statement on line 127 also becomes less convincing once understanding the above. in one investigation (fig2a) the dataset size is manipulated by adding/removing inferred samples while keeping a fixed size of atomic samples. in the other (fig2b), it is manipulated by changing the number of atomic samples. a more likely conclusion, which also can only really be drawn if drawing a figure 2a for multiple sizes of |E| and verifying that it stays the same, is that *percentage of [all inferrable data] that gets into training* is what matters for generalisation. for conclusions on distribution, one would have to hold that percentage (and similarly the percentage of atomic data presented in training) constant, and manipulate only their relative repetitions (which is possible given that both are repeated a lot anyway in this setup).
15. fig 2a: explicitly state |E| in caption/image/somewhere. did you check this for multiple |E| and get the same results?
16. fig 2 caption: at this point in the paper,  test_inferred_id has not been introduced, define things on time
17. line 137 - scaling the data has no effect on relative speed - not very convincing once youve scaled for epochs and not steps (i.e. normalised for dataset size). Unless you mean relative speed between generalisations. Some numbers would be more convincing - define when things have happened and see if ratios hold not just visually - e.g. seems |E|=10k takes twice as long to saturate train than smaller |E|, but not so for test.
18. line 161 S[i,a] - poor phrasing specifically in description of 'a', it is not the input token, it is a marker of input position. I.e. it does not take the value of the actual input tokens (and should not, as then we may have ambiguity because r1 and r2 could even be equal)
19. lines 167-172 do you take care that the perturbed run should be valid, i.e. that the path (h,r1',r2) exists in the graph?
20. 175-176 "the ratio of such alterations (between 0 and 1)" - be more explicit, dont understand
21. line 198 claim that there exists a memorizing circuit: this was not shown, or i missed it. it's a reasonable hypothesis but i dont think it has (or can?) actually been explicitly concluded from anything here.
22. line 208: i havent read ref. [43] in detail, but it seems to me that they say what they find doesnt apply to ADAM, which is what is used here. in fact they mention a 2017 paper suggesting there will not be generalisation with adam. clarify discussion and frame accordingly (is this in agreement or contradiction with previous results)
23. footnote pg 6: too vague, cant get anything from this
24. line 216-217: reasonable hypothesis, unfortunately presented as fact and not verified at all. what if the network is actually using the upper layers to complete the atomic facts? following discussion also not sufficiently hedged as a result.
25. generally: a simple circuit was found for this inference, and using only 2 computation 'steps' in practice (layer 0 to 5, and 5 to 8). if correct, such a finding suggests that most of the layers are redundant. why no experiments on transformers with less layers?
26. in general, consider doing multiple experiments for each task, to see if the same circuits arise every time for these tasks, rather than these just being the solutions two specific models learned.
27. generally: in abstract and in conclusion, the results are said to guide potential improvements to data, training, and architecture. the actual concrete suggestions are spread out through the paper. the paper would be easier to act on if these suggestions were also put together in a clear list at some point in the paper, each with references to its supporting evidence in the paper.
28. section 4: presentation of comparison task general - there is some kind of conflation of facts (things that are true about numbers or about the "underlying data" (for lack of a given term)), samples (tuples that go into the dataset for this underlying data), and rules (how facts and samples relate) here that starts to get fuzzy as the section progresses, tidy up. In particular, for equation 3 to work, there also has to be an understanding of what a sample implies about the underlying data, which is currently missing. (Specifically, the missing part is that there are no 'incorrect' samples. This can be corrected by phrasing eq. 2 with an if-and-only-if instead of only an if (i.e., a double headed arrow), the quantifiers need to be carefully updated for this too.)
29. lines 288-289 took a while to parse, be more explicit
30. lines 294-296 be more explicit about whether talking about atomic or inferred facts
31. line 296 "no surface form clues" - explain/be more explicit
32. 297-298 "unlike most conventienal QA benchmarks where.." - do you have any support for this?
33. 310 "model tends to give up ..." - would appreciate examples and numbers. how did you conclude it was doing this? i want to be convinced too
34. the suggestion on parameter sharing following the results on composition is reasonable, but not explored beyond that (i.e. no parameter sharing transformer was trained to see if it would generalise better), so this is more like a general hypothesis/thought for future work.
35. the comparison between parametric and non parametric memory has a confounding factor that is not addressed: the transformers trained in this paper are trained both on the specific data *and on the task* at once. Meanwhile the 'non parametric models' they are compared to are trained on general NLP tasks, and given the data and task in context. Effectively, the non parametric memory is being evaluated for how well transformers can utilise it *in a zero shot setting*. It is conceivable that a transformer fine tuned specifically for performing the task on data given in parametric memory would have much better performance, and this would be a fairer evaluation of the potential of parametric memory.

**Limitations:**

concluded guidelines on data/training/architecture settings not sufficiently supported, comparison to non parametric memory unfair

---

> ### Author Rebuttal · Authors · 2024-08-05
>
> Thank you for the very informative and constructive comments! We will incorporate them to improve our work in the revised version. Overall, we believe that all the major technical concerns raised here are due to some misunderstandings of our paper, and we will group your comments into different topics and respond in detail below.
>
> **[Comparing non-parametric and parametric memory - Question 1,35]** Our setting on comparing non-parametric memory and parametric memory is very different from comparing in-context learning and fine-tuning, and we believe our experimental setting is far from being “unfair”. To explain in more detail:
> - First, both the grokked model and the LLMs understand the rules in the task and the task objective. This is clear for the grokked model and also true for the strong LLMs we tested given the simplicity of the rules and task objectives involved (we also intentionally made the natural language templates very simple s.t. LLMs have no difficulty in understanding them; see lines 299-300 and Appendix F). We also confirmed in our preliminary experiments by testing the LLMs on easy small-scale instances of the task, where they succeed in no doubt. Based on this, what we are comparing here is solely the capabilities of the different models to make difficult deductions (as those in our test set) based on the given facts.
> - Now, it is important to note that we are **not including any examples constructed in the same way as the test examples in the given facts** (see lines 287-293). In other words, **both** the grokked model and the LLM are doing zero-shot prediction here (which is perfectly fine given they both understand the rules and task objective). We believe the confusion here mainly comes from the given facts (specifically, the ID-ID comparisons as “train_inferred_id” in earlier sections) also serving the role of teaching the rules and task objective to the transformer. A clearer setting would be to take a pretrained (grokked) transformer (e.g., one from Section 4) and then continue training it on the given facts here, however, this won’t make qualitative differences since the given facts can teach the model these things by themselves anyways.
> - Overall, we admit that it is very hard to be perfectly fair, especially given that we don’t have control over the LLM’s training data, but still, we believe our experimental settings are far from being “unfair”.
>
> In terms of terminology: thank you for the suggestion and we will improve the terminologies used in the revised version. It is important to note that prior literature (e.g., [1,2] cited below at the end of the response) has used terms including “non-parametric” and “memory” for the “in-context knowledge” you mentioned, and we are not inventing completely new terms here in the paper.
>
> **[Experiments on models with parameter sharing - Question 34, also in “Strengths”]** We **did** perform experiments on the parameter-shared transformer model (akin to Universal Transformers), and showed that it can achieve highly non-trivial systematic generalization for the composition task. The relevant details are included in Appendix E.2 and mentioned in lines 226-228 in the main content. We believe this is a firm initial step showcasing the potential of the suggested directions implied by our findings and analysis.
>
> **[Ratio between inferred & atomic facts, and repetitions of facts during training - Question 8,12,13,14,17]**
> - We are in the setting where each example in the training set is unique, and we train the model for a very large number of steps/epochs (basically training the model forever). Yes, increasing the ratio (while holding the atomic facts fixed) means that the dataset will be larger (we also mentioned this in line 128), and your calculations on the number of facts are correct. We are not adding these explicitly in the paper due to space limits and will add them in the revised version.
> - Thank you for the suggestion on using percentage instead of ratio - indeed, this could help avoid quantities specific to the dataset construction, and we will consider switching to it in the revised version. Nevertheless, in our setting, the percentage of included inferred facts is always the ratio divided by the KG’s outgoing degree (20), and hence they are only different in units. In other words, it is not that there is “something else that is being directly varied” (which may suggest the existence of unconsidered confounding factors to some degree); we are directly varying it in the first place.
> - Regarding repetitions of facts - having different amounts of repetitions of the facts is another interesting factor to study, but also goes beyond our scope here. Regarding your comment on the necessity of changing the relative repetitions of the facts while holding the percentage fixed (Q13, 14), here we are **not restricting ourselves to comparing different distributions over the same support** - rather, we are talking about **variations in characteristics of the distributions (which could have different supports)**. We believe this should resolve your confusion here. Also, it should be clear at this point that your “more likely conclusion” mentioned in Q14 is really equivalent to our conclusion in the first place.
> - By “relative speed” (Q17), we mean the relative speed of improvements in generalization and training (line 137) for a dataset. So it is valid to rescale the steps (including ones based on dataset size as we did). We believe the visual evidence should already be clear enough, but will consider adding quantitative scores in the revised version.
> - Overall, we believe that your major technical concerns here are mostly caused by misunderstandings potentially due to the writing being a bit abstract at certain places (mostly due to space limit), and we don’t see real pitfalls in our findings or analysis. We will improve the writing and incorporate your suggestions in the revised version.
>
> (continued in comment "Part 2")

---

> ### Author Response · Authors · 2024-08-06
> **Rebuttal by Authors (Part 2)**
>
> **[Tokenizations & rigid input structures - Weakness 1, Question 9]** We actually explicitly mentioned the tokenization scheme in lines 103-104, which immediately implies the input representations. Additionally, we investigated alternative tokenizations in Appendix C to ensure the robustness of our findings to tokenizations. We also performed other preliminary experiments (e.g., different templates) not included in the paper where the number of tokens is larger, and find the results to be consistent. We will add relevant content in the revised version. On the other hand, since we are training from scratch for full control and rigor of the results, the model does not have any pretrained language understanding and hence the inputs we use are indeed (and need to be) quite rigid without natural language variations (like those in real corpora). We do admit that this is a limitation (Appendix H), but note that since we have rather poor understandings even for the simple settings, we believe our work lays a solid foundation for future work to build on and gradually move to more realistic (non-rigid) settings.
>
> **[Overlaiming]** Indeed we are only studying two reasoning types in synthetic settings, but on the other hand, our evaluations are very clean and rigorous, and we believe our experiments with the complex reasoning task (Section 5) where the grokked transformer is shown capable of performing very difficult reasoning are arguably strong and exciting findings. We will consider adjusting the title to a less strong one.
>
> **[Other technical questions]** We respond to other relatively minor technical questions here.
> - Q5: In some sense this is more of a problem of definition; but generally we feel OOD generalization (or, systematicity to be specific) is a very desirable property for models to have but not a requirement for saying that the models can reason.
> - Q10: Indeed the KG here is an abstract one and doesn’t reflect such constraints. One future direction is to make the setting more realistic by adding more natural variations and constraints into the setting.
> - Q15: Thank you for the suggestion. Here |E| is 2000 as in the earlier setting. We did perform experiments across different |E| and found the results to be consistent (Figure 2 in added PDF). We will include these in the revised version.
> - Q18: The second coordinate here could be regarded as the marker for position but it is also the abstract variable name for the corresponding token, which could be more informative and easier to conceptualize. Indeed it obviously should not be the actual token as you mentioned, and we will fix the writing in the revised version.
> - Q19: Yes, we do take care of this, and also made sure to choose from the perturbed examples that end with a different tail entity.
> - Q20: Here we mean the ratio of examples (from the 300 random examples we studied) where the target prediction is altered. We will make this explicit in the revised version.
> - Q21: Indeed we are not able to explicitly show the evidence for the memorizing circuits (which is rather difficult and may require new analysis techniques), but we have convincing evidence for such a hypothesis (lines 188-196 and Appendix D). To reiterate here, the causal strength between S[5, r1] (which is the state that encodes the bridge entity throughout grokking, Figure 4c) and the prediction is very weak at the beginning of grokking (Figure 9a), and grew significantly during grokking (Figure 4b). This suggests that in the beginning, the model is very likely not traversing through the bridge entity when predicting the tail, and hence directly associating h, r1, r2 with t, which is our definition of “memorization”.
> - Q22: Here our purpose is to connect with prior work on implicit regularizations, which could be a potential direction for explaining our observations. We are not trying to be very technically solid here, since even though there is a wide range of prior results along this line, they are all in somewhat restricted settings and no general mathematically proven theory of implicit regularization for deep networks exists to our best knowledge. The specifically cited work in our paper also mostly focuses on linear and restricted ReLU network settings, which are very different from multi-layer transformers. Again, we don’t think deep technical discussion is necessary here for our purpose, and our work could potentially inspire further theoretical investigations along this line.
> - Q24: We did compute the logit lens results for the OOD setting, and found that S[5, r1] encodes the bridge entity (MRR 0.98) and S[5, r2] encodes the second relation (MRR 1.0) as in the ID case, which are strong indications that the lower layers in the OOD setting are performing similar roles as in the ID setting, including storing the OOD atomic facts. We will include these results and expand the discussion in the revised version.
>
> (continued in comment "Part 3")

---

> ### Author Response · Authors · 2024-08-06
> **Rebuttal by Authors (Part 3)**
>
> - Q25: It’s not accurate to conclude “many layers are redundant” from our findings, since it is not the general case that one layer can perform one step, e.g., many times it may take multiple layers to perform one step. Investigating whether we could actually compress the computations into fewer layers is interesting future work.
> - Q26: We did confirm that the same circuits arise across the different settings we experimented with. We will add relevant details in the revised version.
> - Q28: We believe it is implied from our writing (e.g., Section 2) that all the given examples/facts are correct ones. We will make this explicit in the revised version.
> - Q31,32: Here we mean that the query itself does not leak information about the ground-truth proof structure. For conventional QA benchmarks such as NaturalQuestions, TriviaQA, HotpotQA, MuSiQue, etc., the ground truth proof structure can mostly be obtained already by directly parsing/decomposing the query, which is not the case for our constructed task.
> - Q33: We found that with CoT, 70.7% of Gemini’s responses ended up saying that the answer cannot be decided (which we treat as wrong since the answer can be decided). It drops a bit to 58.7% when augmenting with retrieval. One typical example of such cases is included in Table 1 in the added PDF. We will add more discussions and examples in the revised version.
>
> **[Suggestions for writing - Question 2,3,4,6,7,11,16,23,27,29,30]** Thank you for pointing out these issues and suggestions for our writing. These are greatly helpful for improving the paper. We do admit that our current draft is overall a bit abstract and lacks some concrete illustrations, but this is mainly due to the limited space where we need to make the language compact. We still had to put a lot of experiments and details and discussion of limitations/related work in the Appendix, which may have also caused certain confusions and misunderstandings above. These could be rather easily resolved (and we will) in the revised version.
>
> ====
>
> Citations
>
> [1] Zhong et al. Training Language Models with Memory Augmentation. EMNLP-22.
>
> [2] Min et al. Nonparametric Masked Language Modeling. Findings of ACL-23.

---

### Author Rebuttal · Authors · 2024-08-06

We thank all the reviewers for the constructive comments. We have provided detailed responses individually, with additional figures/tables (referred to as the "added PDF") attached here. There are two recurring topics across the reviews which we would like to briefly reiterate here.

**[Synthetic nature of the study and connection to practice]** One clear limitation of our work is its synthetic and abstract nature (highly structured inputs, single-task setting, etc.), which we also admit and discuss in Limitations (Appendix H). Despite our further efforts on ensuring that the results are robust to different setups (Appendix B,C,E), there are still substantial distances from our settings to those in practice. However, we would like to emphasize that our current understandings of the models and their behaviors are very limited even for the idealistic settings we consider here, and we believe our findings and analyses in this work are highly non-trivial and lay a good foundation for future research to build upon, expand, and transition toward more realistic settings.

**[Abstractness of writing]** We admit that our current draft is overall a bit abstract and lacks some concrete illustrations, where it may take multiple passes or some time to think between the words at different places. This is primarily due to the limited space where we need to make the language compact and also move lots of experiments and details to the Appendix, which may have also caused certain confusions and misunderstandings. We believe that these could be easily resolved (and we will) in the revised version.

We look forward to further discussions!

---

> ### Comment · Reviewer_EZN5 · 2024-08-10
> **Comment on the Synthetic nature of the study and connection to practice**
>
> I would like to add that I strongly agree that even though the work is synthetic in nature, this is a fair limitation (not weakness) for a paper to have as long as it's discussed, which the authors do. In a common pursuit of trying to understand how complex models work, controlling the setup to investigate phenomena is a very important part of research that is complementary to approaches that look at the often more complicated and fuzzy realistic non-synthetic tasks and models. The authors do that in a way that yields testable predictions, which they confirm in supplementary experiments in the appendix (e.g. higher weight decay makes grokking happen faster, and this is because the generalising circuit is more efficient). All in all, I believe this paper gives important insight into how transformers learn to do a type of reasoning.

---

> ### Author Response · Authors · 2024-08-14
>
> Thank you very much for your appreciation of our work and for other constructive comments earlier!  In our revised version, we will make things clearer based on feedback from all reviews. We believe our work will inspire more future work in this space, and thank all reviewers again for their time and effort reviewing our work!

---

### Decision · Program_Chairs · 2024-09-25

**Decision:**

Accept (poster)

**Comment:**

The paper investigates the reasoning capabilities of transformers in two synthetic scenarios, measuring compositionality and the ability to compare items. The main takeaway is that reasoning capabilities on unseen cases (generalization) are more evident when the transformers enter the "grokking" phenomenon. Furthermore, they highlight how compositionality is harder than comparing.

Reviewers appreciated the paper's direction (one reviewer was enthusiastic and gave 8, but it's an outlier), but highlighted also a number of shortcomings such as overclaiming results (starting from the title and continuing with the tone of writing thorough the paper) and the limited experimental setting (no ablations). As it is, despite the positive average score, the paper is borderline. I still believe that it can add an interesting perspective to the conference, as it allows to quantify reasoning in some (limited) reasoning scenarios where one can check for latent confounders.

I recommend acceptance *conditioned* on the fact that authors include all reviewers' feedback as promised, especially the one from reviewer jEH8, starting by amending the title.